# PreferThinker: Reasoning-based Personalized Image Preference Assessment

Shengqi Xu♠, Xinpeng Zhou♣, Yabo Zhang♣, Ming Liu♣,✉, Tao Liang♦, Tianyu Zhang♦
Yalong Bai♦, Zuxuan Wu♠, Wangmeng Zuo♣

♠ Fudan University     ♣ Harbin Institute of Technology     ♦ iN2X

**Project Page**

## Abstract

Personalized image preference assessment aims to evaluate an individual user's image preferences by relying only on a small set of reference images as prior information. Existing methods mainly focus on general preference assessment, training models with large-scale data to tackle well-defined tasks such as text-image alignment and aesthetics. However, these approaches struggle to handle personalized preference assessment because user-specific data are typically scarce and not easily scalable, and individual tastes are often diverse and complex. To overcome these challenges, we introduce a common preference profile that serves as a bridge across users, allowing large-scale user data to be leveraged for training profile prediction and capturing complex personalized preferences. Building on this idea, we propose a reasoning-based personalized image preference assessment framework that follows a *predict-then-assess* paradigm: it first predicts a user's preference profile from reference images, and then provides interpretable, multi-dimensional scores and assessments of candidate images based on the predicted preference profile. To support this, we first construct a large-scale Chain-of-Thought (CoT)-style personalized assessment dataset annotated with diverse user preference profiles and high-quality CoT-style reasoning, enabling explicit supervision of structured reasoning. Next, we adopt a two-stage training strategy: a cold-start supervised fine-tuning phase to empower the model with structured reasoning capabilities, followed by reinforcement learning to incentivize the model to explore more reasonable assessment paths and enhance generalization. Furthermore, we propose a similarity-aware prediction reward to encourage better prediction of the user's preference profile, which facilitates more reasonable assessments exploration. Extensive experiments demonstrate the superiority of the proposed method.

## 1 Introduction

Image preference assessment is essential for both evaluating and aligning generative models with human preferences, playing a key role in applications like content recommendation. However, most existing works (Kirstain et al., 2023; Xu et al., 2023) focus on assessing *general* preferences, such as text-image alignment and aesthetics, relying on large-scale general preference data. The practical but challenging task of *personalized* preference assessment remains largely underexplored, where only a limited amount of personalized data is available for each user as prior information to assist in the assessment. As shown in Fig. 1 (a), personalized preference assessment is particularly challenging for two main reasons. First, unlike *easily scalable* general preference data, where most users share similar assessment criteria, each user's personalized data is typically *limited* and *distinct* from others, making large-scale, user-specific training difficult. Second, personalized preferences are often *complex* and *diverse*, covering multiple dimensions such as art style, color, and art medium, which are more challenging than *clear* general human preferences (*e.g.* text-image alignment and aesthetics).

Current research on image preference assessment mainly falls into two categories: *CLIP-based* methods and *Multimodal Large Language Model (MLLM)-based* methods. *CLIP-based* methods (Wu et al., 2023b;a) typically fine-tune the CLIP model using large-scale text-image pairs with similarity to assess general preferences. However, they struggle with personalized preferences since the amount of personalized data is typically limited for large-scale training. Besides, simple similarity fails to capture complex individual preferences and lacks interpretability, as it outputs only a numerical score.

(a) Challenges of personalized preference *vs.* general preference    (b) Common preference profile for bridging various users

Figure 1: Illustration of challenges and motivation. (a) The general preference data is easily scalable since users share common assessment criteria, while personalized preference data for each user is typically limited and unscalable, as each user's preferences are distinct. Besides, general preferences are often clear (*e.g.* text-image alignment and aesthetics), while personalized preferences are typically complex and diverse. (b) We propose a preference profile comprising multiple common visual elements, based on the observation that although each user's personalized preferences are unique, the key visual elements that shape them are shared and can therefore serve as a bridge to connect users.

*MLLM-based* methods (Wang et al., 2025c; Xu et al., 2024) achieve interpretable preference assessment by fine-tuning the MLLM on large-scale visual question-answering (VQA) pairs. However, the scarcity of personalized images makes it difficult to obtain sufficient VQA pairs for large-scale training, which limits their effectiveness for personalized preference assessment. Overall, most existing methods are designed for general preferences and therefore struggle to handle personalized preference. Recently, an MLLM-based method named ViPer (Salehi et al., 2024) was proposed for personalized preference assessments. It directly feeds both personalized reference images and candidate images into an MLLM and fine-tunes the model using a supervised score regression strategy. Nevertheless, it only *implicitly* utilizes reference images to assist in evaluating the candidate images, failing to fully leverage the critical prior information they contain. Moreover, it lacks interpretable reasoning steps that explain how to produce its final score based on the individual reference images.

To address the above issues, we propose a reasoning-based personalized preference assessment system with preference profile prediction, named **PreferThinker**. Our key insight is to bridge various users by introducing a preference profile comprising multiple visual elements (*e.g.* color, art style), as shown in Fig 1(b). *Although each user's personalized preferences are unique, the fundamental elements that form them are common and can serve as a bridge across users.* This profile offers three main advantages. First, the element-level profile allows us to characterize personalized preferences, which alleviates the challenge of complex individual tastes. Second, since these visual elements can be shared and learned across users, we can leverage large-scale user samples for training to effectively predict each user's visual preference profile, thereby mitigating the issue of limited personalized data. Third, it provides a solid foundation for subsequent interpretable and multi-dimensional assessments.

Specifically, given a user's personalized reference images (preferred and non-preferred) and two candidate images, where reference images may reflect multiple aspects of user's tastes. PreferThinker follows a *predict-then-assess* CoT-style structure for personalized preference assessment. As shown in Fig. 2, the process consists of two stages: 1) Profile Prediction: PreferThinker first predicts the user's visual preference and non-preference profiles comprising multiple visual elements based on the reference images. 2) Multi-dimensional and Interpretable Assessment: Using the predicted profile, PreferThinker provides interpretable scores for the candidate images across multiple dimensions and produces a final result based on the total score. This *predict-then-assess* framework not only enables accurate assessment, but also achieves interpretable scoring grounded in the user's reference images.

To this end, we first construct a large-scale CoT-style personalized image preference assessment dataset, named **PreferImg-CoT**, which contains 60,000 user samples, annotated with preference profiles and high-quality CoT-style assessments, enabling explicit supervision of structured reasoning. Next, we adopt a two-stage training strategy: a cold-start supervised fine-tuning (SFT) phase to elicit the model structured reasoning, followed by reinforcement fine-tuning (RFT) based on Group Relative Policy Optimization (GRPO) to guide model toward exploring more reasonable assessments and enhancing generalization. Moreover, we find that accurate preference profile prediction facilitates exploring reasonable assessments, which motivated us to propose a similarity-aware prediction reward to better predict user's preference profile. Overall, our main contributions are summarized as follows:

- We propose PreferThinker, a reasoning-based personalized image assessment system with preference profile prediction, which achieves accurate assessment along with interpretable and multi-dimensional scoring based on the predicted profile.

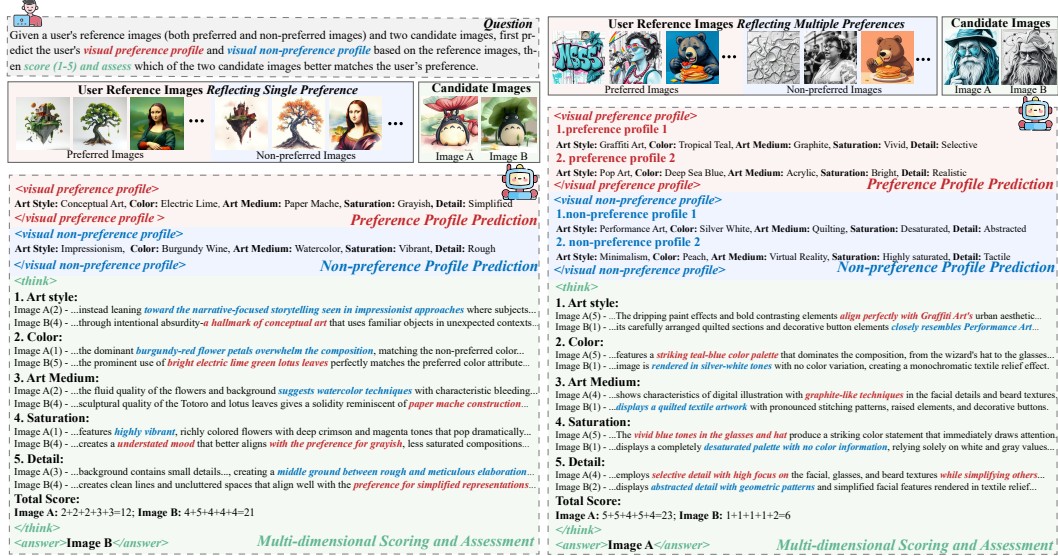

Figure 2: Examples of PreferThinker for personalized image preference assessment. In the *think* stage, *red text* denotes alignment with preference profiles, while *blue text* denotes alignment with non-preference profiles. *See Appendix D for more complete reasoning examples.*

- We construct PreferImg-CoT, the first CoT-style personalized assessment dataset, annotating with preference profile and CoT-style reasoning, facilitating the supervision of model interpretability.

- We adopt a training strategy comprising cold-start and GRPO-based RFT to enables the system to master the reasoning capability. Furthermore, we propose a similarity-aware prediction reward to better predict the user's preference profile, facilitating more reasonable assessments exploration.

- We compare PreferThinker with existing methods on the proposed and existing datasets. Extensive experiments confirm that PreferThinker outperforms SOTA methods. Moreover, the predicted preference profile of PreferThinker can also benefit personalized image generation.

## 2 RELATED WORK

**Image Preference Assessment** is crucial for both evaluating and aligning generative models with human preferences. While metrics such as FID (Heusel et al., 2017) and CLIP scores (Radford et al., 2021) assess image quality and text-image consistency, they are limited in capturing human preferences. To address this, most works (Kirstain et al., 2023; Liang et al., 2024; Li et al., 2024; Zhang et al., 2024; Ba et al., 2025; Chen et al., 2025a) employ large-scale preference data to fine-tune CLIP models for assessing general preferences. However, they lack interpretability, acting as black boxes that output only a score. Recently, MLLM-based methods(Zhou et al., 2025; Wang et al., 2025b; Mo et al., 2025a; Gambashidze et al., 2025) have emerged, benefiting from the powerful multimodal understanding of MLLMs. For example, (Xu et al., 2024) and (Wang et al., 2025b) fine-tune MLLMs using large-scale visual question-answering pairs to achieve interpretable assessments. Nevertheless, most existing methods focus on general preferences and struggle with personalized preferences due to the scarcity of personalized data and complexity of individual tastes. Though (Salehi et al., 2024) stands out by proposing a personalized assessment method, it naively feeds reference and candidate images into an MLLM for score regression, thus lacking interpretability. In this work, we propose a reasoning-based personalized assessment system that achieves both accurate assessments and interpretable scoring.

**Image Preference Assessment Dataset** is essential for training preference assessment models. (Xu et al., 2023) proposes ImageRewardDB, the first dataset for training models to assess general preferences (*e.g.* text-image alignment, aesthetics and safety), containing 137K candidate image pairs. Further, (Kirstain et al., 2023) and (Wu et al., 2023a) respectively developed PickaPic_v2 and HPD_v2, two large-scale and diverse datasets for learning human preferences. However, most existing public datasets focus on training models to capture general preferences, with limited consideration for personalized preferences. While (Salehi et al., 2024) curates a simulated personalized preference dataset, it remains unreleased. In this work, we construct a large-scale personalized assessment dataset annotated with diverse preference profiles and CoT-style reasoning to advance this field.

**Reasoning-based Multimodal Large Language Models** have achieved significant progress in multimodal reasoning abilities through CoT-based or RL-based fune-tuning. Deepseek-R1 (Guo et al., 2025a) employs the GRPO algorithm (Shao et al., 2024) with the rule-based reward to improve reasoning abilities of LLM without critic models. Inspired by this, several works have incorporated GRPO-based post-training into MLLMs to enhance the multimodal reasoning in various tasks, including visual perception (Jiang et al., 2025; Huang et al., 2025; Liu et al., 2025b; Zhang et al., 2025a; Yu et al., 2025; Liu et al., 2025a; You & Wu, 2025) and visual understanding (Chen et al., 2025b; Pan et al., 2025; Ni et al., 2025), math problem solving (Guo et al., 2025b; Zhang et al., 2025b; Li et al., 2025b), and visual quality assessment (Wu et al., 2025; Li et al., 2025a; Cai et al., 2025). Closest to ours, (Wang et al., 2025b) explores reasoning-based image preference assessment through RL-based post-training. However, this method focuses on assessing general preferences. Distinctly, PreferThinker is the first reasoning-based personalized assessment system. Inspired by Deepseek-R1, we adopt a two-stage training strategy comprising cold-start SFT and GRPO-based RFT to enable the system to master the reasoning capability Furthermore, we propose a similarity-aware prediction reward to better predict user's preference profile, facilitating more reasonable evaluations exploration.

## 3 Reasoning-based Personalized Image Preference Assessment

In this work, we propose PreferThinker, a reasoning-based assessment system with preference profile prediction. Given a few individual reference images (both preferred and non-preferred) and two candidate images, PreferThinker assesses the candidate images following a *predict-then-assess* CoT-style structure. As shown in Fig. 2, it first predicts the user's visual preference and non-preference profiles based on the reference images. Next, it utilizes the predicted profiles as criteria to score the candidate images across multiple dimensions in an interpretable manner and produces a final result based on the total score. By leveraging this *predict-then-assess* paradigm, PreferThinker not only achieves accurate assessment but also providing interpretable scoring based on the predicted profiles.

Specifically, we first introduce a preference profile composed of multiple common visual elements to bridge various users (Section 3.1). Next, we construct a large-scale CoT-style dataset annotated with preference profiles and high-quality reasoning to provide reasoning supervision (Section 3.2). We then employ a two-stage training strategy comprising cold-start SFT and RL-based post-training to enable the system to master the reasoning capability (Section 3.3). Moreover, we discover that more accurate profile prediction facilitates exploring more reasonable assessments, which motivates us to propose a similarity-aware prediction reward to better predict preference profile (Section 3.4).

### 3.1 Visual Preference Profile

Most existing methods mainly rely on large-scale, easily scalable general preference data to train models for general preference assessment. However, they struggle to handle personalized preference due to the limited personalized data for each user and user heterogeneity. To address this issue, we propose a preference profile comprising multiple common visual elements to bridge various users. We argue that *although each user's personalized preferences are unique, the key visual elements that shape these preferences are shared and can serve as a intermediate bridge to connect various users*.

To formalize a user's complex personalized visual preferences, we identify 15 visual elements that most frequently appear in Lexica's [1] text prompts and strongly influence user preference toward personalized image generation. We then conduct a user study with 100 participants, asking each to select the five most important visual elements as visual preference profile. The result (See Fig S1 in the Appendix) reveals that *art style*, *color*, *detail*, *art medium* and *saturation* are voted as the most representative visual elements for characterizing personalized preferences. To ensure preference profile diversity, we collect a rich vocabulary of related terms for each visual element, totaling 288 words (See Fig. S2 in the Appendix), laying a solid foundation for constructing a large-scale and diverse dataset for personalized image preference assessment.

### 3.2 Large-scale CoT-style Personalized Preference Assessment Dataset

To enable the model to master CoT-style reasoning capabilities, we construct a large-scale CoT-style dataset that provides high-quality reasoning supervision. Since most existing public datasets focus on *general* preference assessment, we first construct a new dataset called PreferImg, dedicated to

---

[1] https://lexica.art/

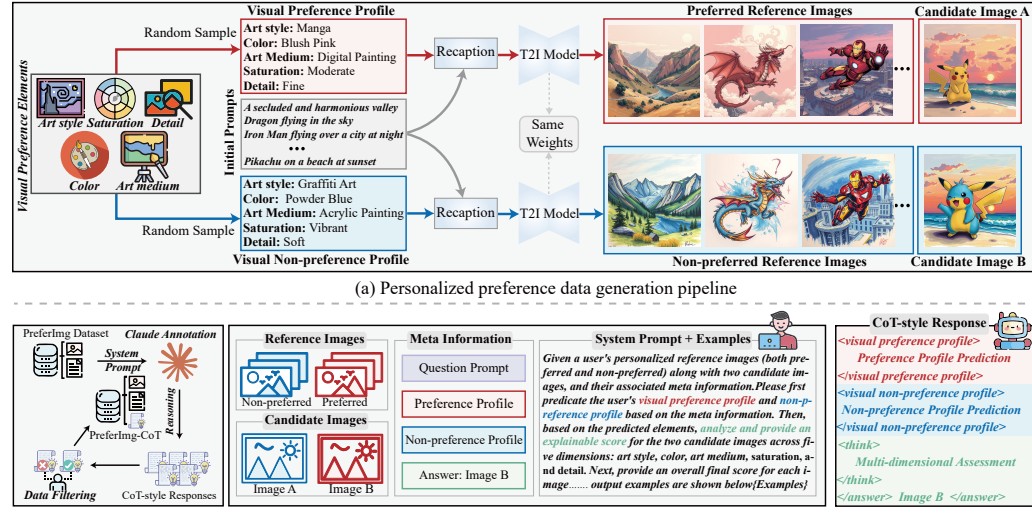

(a) Personalized preference data generation pipeline

(b) CoT-style dataset construction    (c) Multimodal information and system prompt for Claude 3.7    (d) CoT-style response template

Figure 3: Illustration of the proposed dataset PreferImg-CoT. (a) Personalized preference data generation pipeline. (b) Overview of CoT-style dataset construction: Claude annotation and data filtering. (c) Prompt design for Claude 3.7 to generate CoT-style response. (d) CoT-style response template, including preference profiles prediction, multi-dimensional assessment and answer.

*personalized* image preference assessment, annotated with diverse visual preference profiles. We then employ the advanced Claude 3.7 model (Anthropic, 2024) to generate high-quality CoT-style annotations on the proposed PreferImg, ultimately producing our CoT-style dataset, PreferImg-CoT.

**Personalized Preference Assessment Data Generation.** Due to the difficulty in acquiring personalized preference data with annotated preference profiles, such datasets are currently unavailable. To fill this gap, inspired by (Mo et al., 2025b), we construct a large-scale personalized assessment dataset containing 80K simulated users, each with a distinct preference profile. The data generation pipeline is shown in the Fig. 3(a). Specifically, we first randomly sample five visual preference elements to assign visual preference and non-preference profiles to each user. To account for the fact that real users may have multiple personalized preferences, several preference profiles are assigned to a subset of users. Then, we combine these profiles with initial prompts and feed into a text-to-image model to generate each user's reference images (preferred and non-preferred) and two candidate images. To further guarantee content diversity, we follow the PrefGen (Mo et al., 2025b)to select 190 K prompts from Lexica, DiffusionDB (Wang et al., 2022), and COCO (Lin et al., 2014) as initial prompts. Ultimately, we establish a dataset annotated with preference profiles, comprising 80 K users, 20K of whom are associated with multiple preference profiles, along with 1.36 million images, laying a solid foundation for subsequent CoT-style dataset construction.

**CoT-style Dataset Construction.** To construct a high-quality CoT-style dataset, we design a two-stage construction process: Claude annotation and data filtering, as shown in Fig. 3(b). First, we employ Claude 3.7 with prompting to generate CoT style responses. Figure 3(c) shows that the prompt contains four key components: (1) user reference images (both preferred and non-preferred), (2) two candidate images, (3) meta-information including the question prompt, visual preferences and non-preferences, and the correct answer, and (4) system prompt along with correct response examples. Based on this prompt, Claude 3.7 follows a *predict-then-assess* structure. It first predicts the user's visual preference profile from the reference images, which serves as a criterion for assessment. It then generates a CoT-style analysis that includes multi-dimensional, interpretable scoring and assessments for the candidate images, and finally outputs the final result. The response template is shown in Fig. 3(d). To ensure annotation quality, we further filter out any illogical or incorrect CoT-style responses, such as those with inconsistent reasoning or a mismatch between the prediction and ground-truth. Finally, we curate PreferImg-CoT, a large-scale CoT-style dataset with 60,000 user samples based on the proposed PreferImg dataset, which serves as the foundation for our subsequent cold-start SFT.

## 3.3 TRAINING STRATEGY

We employ a two-stage training strategy to elicit and incentivize the model's structured reasoning capabilities. First, we conduct a cold-start supervised fine-tuning (SFT) phase to teach the model how

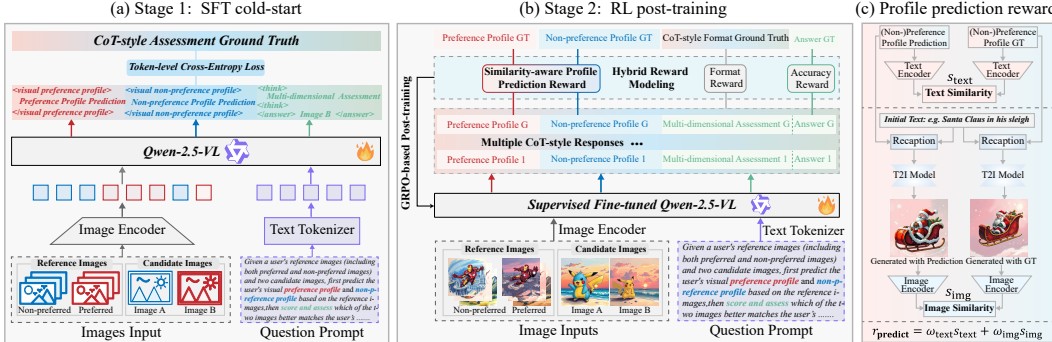

Figure 4: Illustration of training strategy and proposed prediction reward: (a) Cold-start SFT to teach structured reasoning; (b) RL-based post-training to explore more reasonable assessments and enhance model generalization. (c) Similarity-aware prediction reward for better preference profile prediction.

to perform structured reasoning. Then, we employ reinforcement learning (RL)-based post-training to encourage the model to explore more reasonable assessment paths and enhance its generalization.

**Stage 1: Supervised Fine-tuning for Cold-start Initialization.** We utilize Qwen2.5-VL-7B (Bai et al., 2025) as the baseline model and conduct a SFT with the proposed CoT-style dataset PreferImg-CoT, as shown in Fig. 4(a). During the cold-start phase, the model learns to first predict the user's preference profiles based on reference images and then leverages the predicted profiles as criteria to produce interpretable, multi-dimensional scores and assessments for candidate images. We train the model using a standard autoregressive language modeling objective and apply a token-level cross-entropy loss to provide strong supervision throughout the generation process:

$$\mathcal{L}_{SFT}(\theta) := -\mathbb{E}_{(x,y)\sim\mathcal{D}_{\text{CoT}}} \sum_{t=1}^{T} \log P\left(y_t \mid x, y_{<t}; \theta\right), \tag{1}$$

where $\mathcal{D}_{\text{CoT}}$ is our PreferImg-CoT, and $(x, y)$ is the input query and CoT-style target response.

**Stage 2: GRPO-based Reinforcement Learning for Post-training.** Although cold-start endows the model with structured reasoning capabilities, its supervised training paradigm limits the exploration of diverse reasoning paths and restricts generalizability. To explore more reasonable assessments and enhance generalizability, we employ the Group Relative Policy Optimization (GRPO)-based reinforcement learning for exploration-driven post-training, as shown in Fig. 4(b). Unlike traditional policy optimization methods (Schulman et al., 2017), GRPO efficiently generates diverse reasoning responses by optimizing policy gradients on a sample group without requiring a critic model.

*Group Relative Policy Optimization.* The MLLM generates a group of $G$ CoT-style outputs $\{o_1, o_2, \ldots, o_G\}$ for each input query $x$ from current policy $\pi_\theta$. Each output contains preference profiles prediction, multi-dimensional assessment and a final answer. For each $o_i$, we compute a scalar reward $r_i$, and normalize these rewards to estimate its group-relative advantage $A_i$:

$$A_i = \frac{r_i - \max\left(\{r_1, r_2, \cdots, r_G\}\right)}{\text{std}\left(\{r_1, r_2, \cdots, r_G\}\right)} \tag{2}$$

Then, the GRPO training objective is as follows:

$$\mathcal{J}_{GRPO}(\theta) = \mathbb{E}_{x\sim P(X), \{o_i\}_{i=1}^G \sim \pi_{\theta_{\text{old}}}(O|x)} \left[\frac{1}{G} \sum_{i=1}^{G} \min\left(\frac{\pi_\theta(o_i \mid x)}{\pi_{\theta_{\text{old}}}(o_i \mid x)} A_i, \right. \right. \tag{3}$$

$$\left. \left. \text{clip}\left(\frac{\pi_\theta(o_i \mid x)}{\pi_{\theta_{\text{old}}}(o_i \mid x)}, 1 - \varepsilon, 1 + \varepsilon\right) A_i\right) - \beta \mathbb{D}_{KL}(\pi_\theta \| \pi_{\text{SFT}})\right]$$

where $\varepsilon$ and $\beta$ are hyperparameters, and $\pi_{\text{SFT}}$, $\pi_\theta$, and $\pi_{\theta_{\text{old}}}$ are the model after cold-start SFT, the optimized model and the old policy model. The KL divergence term $\mathbb{D}_{KL}(\pi_\theta \| \pi_{\text{SFT}})$ promotes controlled policy updates, balancing exploration and stability throughout training.

### 3.4 SIMILARITY-AWARE PREFERENCE PROFILE PREDICTION REWARD

We find that accurate profile prediction facilitates the exploration of reasonable assessments, motivating us to propose a prediction reward to improve the accuracy of profile predictions. Figure 5

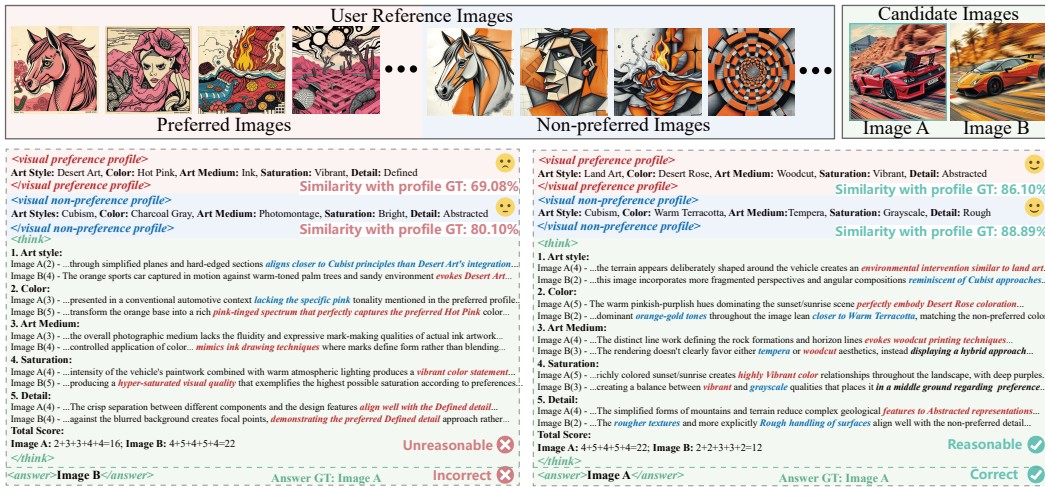

Figure 5: Effectiveness of the proposed prediction reward (PR). (a) The model without PR predicts profiles differed from GT, leading to unreasonable assessment and incorrect answer. (b) The model with PR predict the profiles closer to GT and provide more reasonable assessment with correct answer.

illustrates a comparison of personalized assessment cases with and without the prediction reward . The results show that without the reward, the predicted profiles differ from the ground-truth (GT), leading to unreasonable assessments and an incorrect answer. Conversely, the model with the reward predicts the profiles closer to GT and provide more reasonable assessments with a correct answer.

**Similarity-aware Prediction Reward Design.** To incentivize the model to predict the user's preference profile more accurately, we propose a *text-image* similarity-aware prediction reward in Fig. 4(c). Given a user's personalized reference images, PreferThinker first predicts the user's visual preference profile $\hat{V}_+$ and non-preference profile $\hat{V}_-$. First, we measure the prediction accuracy via computing the *text* semantic similarity $s_{\text{text}}$ between the predicted and ground-truth profiles:

$$s_{\text{text}} = Sim_{\text{text}}(\hat{V}_+, V_+) + Sim_{\text{text}}(\hat{V}_-, V_-), \tag{4}$$

where $Sim_{\text{text}}$ measures the similarity between two texts, $V_+$ and $V_-$ are ground-truth preference and non-preference profiles. We use SBERT (Reimers & Gurevych, 2019) to compute the text similarity.

Second, considering that image and text are the two sides of the same coin and should be tightly coupled. Therefore, we also measure the prediction accuracy via computing the *image* similarity $s_{\text{img}}$ between the generated images based on the predicted and ground-truth visual preference profiles. Specifically, given an initial text $T_{\text{initial}}$, we recap it by incorporating the predicted and ground-truth visual preference profiles, and then generate images using T2I model (Labs, 2024):

$$\begin{aligned} \hat{I}_+ = T2I(R(T_{\text{initial}}, \hat{V}_+)), \quad I_+ = T2I(R(T_{\text{initial}}, V_+)), \\ \hat{I}_- = T2I(R(T_{\text{initial}}, \hat{V}_-)), \quad I_- = T2I(R(T_{\text{initial}}, V_-)), \end{aligned} \tag{5}$$

where $T2I$ is the text-to-image model, $R$ denotes the recaption process. Then, we compute the image similarity between the images generated with predictions and ground-truth:

$$s_{\text{img}} = Sim_{\text{img}}(\hat{I}_+, I_+) + Sim_{\text{img}}(\hat{I}_-, I_-), \tag{6}$$

where $Sim_{\text{img}}$ measures the similarity between two images. We use DreamSim (Fu et al., 2023) to compute the image similarity. Finally, the profile prediction reward function is a weighted combination of two similarity measures, designed to balance their contributions:

$$r_{\textbf{predict}} = w_{\text{img}} s_{\text{img}} + w_{\text{text}} s_{\text{text}}, \tag{7}$$

where $w_{\text{img}}$ and $w_{\text{text}}$ are hyperparameters that balance the contribution of each similarity.

**Hybrid Reward Modeling.** Apart from considering the prediction reward $r_{\textbf{predict}}$, we utilize a format correct reward $r_{\textbf{format}}$ and an assessment accuracy reward $r_{\textbf{accuracy}}$ to encourage the model to produce the correct response format and assessment results:

| Preferred reference images | Predicted preference profile | Initial image | Personalized image | Initial image | Personalized image |
|---|---|---|---|---|---|

*Art style:* Naive art
*Color:* Coral pink
*Art medium:* Paper mache
*Saturation:* Vibrant
*Detail:* Textured

"Starry night"          "A smiling cyberpunk girl"

Figure 6: Personalized image generation with the predicated preference profile.

- *Format Correctness Reward* ($r_{\mathbf{format}}$). The format reward $r_{\mathbf{format}}$ is awarded 1 when the response follows the required structure: 1) the predicted preference and non-preference profiles are enclosed within the `<visual preference profile>` and `<visual non-preference profile>` tags, respectively, 2) the multi-dimensional assessment is placed in the `<think>` tags, 3) the final result is contained in the `<answer>` tags; otherwise, the $r_{\mathbf{format}}$ is set to 0.

- *Assessment Accuracy Reward* ($r_{\mathbf{accuracy}}$). Given the ground truth of Assessment result $E$ and the produced assessment result $\hat{E}$. The reward $r_{\mathbf{accuracy}}$ is set to 1 when $\hat{E} = E$, and 0 otherwise.

Furthermore, during the RL-based post-training, we utilize a weighted combination to balance these reward functions. The total reward $r$ for a response is weighted sum of the three rewards:

$$r = w_p r_{\mathbf{predict}} + w_f r_{\mathbf{format}} + w_a r_{\mathbf{accuracy}} \tag{8}$$

where $w_p$, $w_f$ and $w_a$ are hyperparameters that balance the contribution of each reward.

# 4 EXPERIMENTAL RESULTS

## 4.1 EXPERIMENTAL SETTINGS

**Datasets.** We conduct experiments on the proposed PreferImg and real user dataset PickaPic (Kirstain et al., 2023). Based on the PreferImg, we construct a benchmark of 1,500 users, categorizing them into single-preference (SP) and multi-preference (MP) groups based on whether reference images reflect singular or multiple preferences. To evaluate generalization, we further divide the benchmark into seen and unseen subsets based on whether users' profiles are included in the training data. Since no real-user dataset exists for personalized preference assessment, we process PickaPic, originally collected from real user interactions for *general preference assessment*, by grouping samples by user IDs, resulting in a user-specific benchmark with 894 user samples. *Note that although the samples are grouped by users, the assessment labels still reflect general preferences*, unlike PreferImg, which targets personalized preferences.

Table 1: Quantitative comparison of assessment accuracy with existing methods on the proposed PreferImg and real user dataset PickaPic. SP denotes that the user's reference image reflects single preference, while MP is multiple preferences. † denotes the dataset's labels are for *general preference assessment*. ∗ denotes the methods that support reference images as input for in-context learning. The **best** and **second-best** performances are highlighted.

| Method | #Param | Seen PreferImg (SP) | Seen PreferImg (MP) | Unseen PreferImg (SP) | Unseen PreferImg (MP) | PickaPic† | Avg |
|---|---|---|---|---|---|---|---|
| *CLIP-based Models* | | | | | | | |
| PickScore [NIPS2023] | 986M | 49.6 | 48.4 | 51.2 | 56.4 | **67.9** | 54.7 |
| ImageReward [NIPS2023] | 478M | 52.2 | 51.2 | 49.0 | 57.2 | 60.7 | 54.1 |
| HPSv2 [ICCV2023] | 2B | 51.4 | 50.0 | 50.8 | 57.6 | 63.3 | 54.6 |
| CLIPScore [ICML2023] | 428M | 50.2 | 54.0 | 52.6 | 52.0 | 58.0 | 53.4 |
| Aesthetics [arXiv2021] | 304M | 50.6 | 48.0 | 48.6 | 50.8 | 48.6 | 49.3 |
| CycleReward [ICCV2025] | 477M | 48.0 | 50.8 | 50.0 | 56.0 | 62.6 | 53.5 |
| *MLLM-based Models* | | | | | | | |
| UnifiedReward [arXiv2025] | 7B | 49.0 | 46.0 | 50.6 | 57.2 | 61.7 | 52.9 |
| UnifiedReward-Think [NIPS2025] | 7B | 48.6 | 47.2 | 48.0 | 52.8 | 60.5 | 51.4 |
| LLaVA-Reward [ICCV2025] | 8.2B | 53.2 | 51.6 | 52.4 | 58.4 | 62.2 | 55.6 |
| ViPer∗ [ECCV2024] | 8B | 92.4 | 78.0 | **93.4** | 80.0 | 62.2 | 81.2 |
| *Open-Source MLLMs* | | | | | | | |
| Qwen2.5-VL-7B (*Base*)∗ | 7B | 75.4 | 62.0 | 72.0 | 64.8 | 58.0 | 66.4 |
| InternVL-3.5-8B∗ | 8B | 80.0 | 64.0 | 80.6 | 65.6 | 56.0 | 69.2 |
| GLM-4.1V-9B-Thinking∗ | 9B | 86.8 | 73.2 | 86.8 | 72.0 | 62.8 | 76.3 |
| *Closed-Source MLLMs* | | | | | | | |
| Claude 3.7∗ | - | 93.8 | **83.2** | 90.2 | **86.0** | 64.9 | **83.6** |
| OpenAI-GPT-4o∗ | - | **94.2** | 80.4 | 92.2 | 85.2 | **65.7** | 83.5 |
| Doubao-1.5-vision-pro-32k∗ | - | 92.2 | 78.4 | 90.0 | 76.4 | 63.8 | 80.2 |
| Gemini-2.5-flash∗ | - | 85.2 | 61.6 | 83.6 | 67.2 | 61.1 | 71.7 |
| *Ours* | | | | | | | |
| **PreferThinker**∗ [This paper] | 7B | **96.6** | **92.0** | **96.4** | **92.8** | **65.7** | **88.7** |
| *vs. prev. SoTA* | - | +2.4 | +8.8 | +3.0 | +6.8 | -2.2 | +5.1 |
| *vs. Base Model* | - | +21.2 | +30.0 | +24.4 | +28.0 | +7.7 | +22.3 |

**Implementation Details.** PreferThinker is initialized from Qwen2.5-VL-7B and trained on a cluster of 8 NVIDIA A100 GPUs. We utilize the LLaMA-Factory (Zheng et al., 2024) for cold-start SFT, and train for only one epoch to prevent overfitting. After the SFT, we utilize the VLM-R1 framework (Shen et al., 2025) for RL-based post-training. We set a learning rate of 1e-5 with 6 rollout samples per input. The reward weights $w_p$, $w_f$ and $w_a$ are set to 0.7, 0.3 and 1.0. See Appendix A for details.

**Baselines.** We compare PreferThinker with (1) CLIP-based models: PickScore (Kirstain et al., 2023), ImageReward (Xu et al., 2023), HPSv2 (Wu et al., 2023a), CLIPScore (Radford et al., 2021), Aesthetics (Schuhmann et al., 2021), CycleReward (Bahng et al., 2025); (2) MLLM-based methods:, UnifiedReward (Wang et al., 2025c) UnifiedReward-Think (Wang et al., 2025b), LLaVA-Reward

Table 2: Ablation study of PreferThinker. *Ass.*: Assessment accuracy. *Pred.*: Profile prediction accuracy.

| Base | SFT | RL | PR | Seen-SP | | Unseen-SP | | Seen-MP | | Unseen-MP | |
|------|-----|----|----|---------|---------|-----------|---------|---------|---------|-----------|---------|
| | | | | *Ass.* | *Pred.* | *Ass.* | *Pred.* | *Ass.* | *Pred.* | *Ass.* | *Pred.* |
| ✓ | | | | 75.4 | 70.4 | 72.0 | 70.88 | 62.0 | 71.1 | 64.8 | 71.1 |
| ✓ | ✓ | | | 92.0 | 84.2 | 91.8 | 85.3 | 81.2 | 73.9 | 81.6 | 74.2 |
| ✓ | | ✓ | | 89.6 | 79.6 | 91.0 | 80.3 | 78.4 | 76.9 | 77.2 | 73.5 |
| ✓ | ✓ | ✓ | | 94.9 | 84.1 | 95.2 | 85.2 | 90.4 | 72.5 | 89.2 | 74.1 |
| ✓ | ✓ | ✓ | ✓ | 96.6 | 85.4 | 96.4 | 86.2 | 92.0 | 78.6 | 92.8 | 80.7 |

Table 3: Performance and speed of PreferThinker without *explicit* CoT.

| Method | *Accuracy↑* | | *Speed↓* |
|--------|-----------|---------|---------|
| | PreferImg | PickaPic[†] | |
| ViPer | 86.0 | 62.2 | **2.19 s** |
| OpenAI-GPT-4o | 88.0 | **65.7** | 12.92 s |
| Claude 3.7 | **88.3** | 64.9 | 18.22 s |
| Doubao-1.5-vision | 84.3 | 63.8 | 5.39 s |
| **Ours (w/o CoT)** | **93.2** | **65.2** | **0.80 s** |

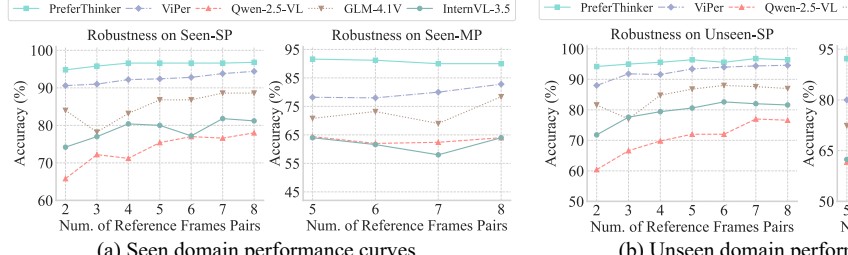

(a) Seen domain performance curves          (b) Unseen domain performance curves

Figure 7: Robustness to number of personalized reference images on PreferImg.

(Zhou et al., 2025), and ViPer (Salehi et al., 2024); (3) Open-source MLLMs: Qwen2.5-VL-7B (Bai et al., 2025), InternVL-3.5 (Wang et al., 2025a), GLM-4.1V-9B (Hong et al., 2025); (4) Closed-source MLLMs: Claude 3.7, GPT-4o, Doubao-1.5-vision, and Gemini-2.5. For a fair comparison, we categorize the methods based on whether support reference images as input for in-context learning.

## 4.2 COMPARISON RESULTS

**Comparison on Seen User Data.** In Table 1, we compare PreferThinker with existing methods on the seen user data of PreferImg. It is observed that CLIP-based and most MLLM-based methods perform poorly, since they lack modeling for personalized preferences. Though ViPer achieves relatively good results on the SP data, its accuracy decreases when handling the MP data, and it lacks interpretability as it only outputs a numerical score. Further comparisons with the open-MLLMs reveal that directly utilizing middle-sized models struggle with personalized preference assessment. In contrast, closed-source models achieve comparable performance, thanks to their powerful prior knowledge. Overall, PreferThinker achieves the highest accuracy and provides interpretable assessments and scores.

**Comparison on Unseen User Data.** To validate generalization, we compare performance on unseen users in Table 1. PreferThinker maintains the highest accuracy on PreferImg, revealing strong generalization. However, nearly all methods perform poorly on the real user benchmark, PickaPic, due to its complexity. Though we group PickaPic by user ID and assign user-specific reference images, *the labels still reflect general preferences*. Thus, methods designed for general preferences perform better on PickaPic. PickScore achieves the best performance since PickaPic serves as a seen dataset for it, while our method ranks second on unseen PickaPic data, further confirming its generalizability. *Please refer to Appendix D.3 for CoT-style assessment visualization of real users.*

## 4.3 ABLATION STUDY AND DISCUSSION

**Effectiveness of Cold-start SFT and RL Post-training.** Table 2 demonstrates the critical roles of both cold SFT and RL training phase. Removing either phase leads to a notable degradation in model performance, confirming their importance for accurate preference assessment. Moreover, the use of RL after SFT significantly improves the model's performance on unseen user data, particularly for multi-preference (MP) users, revealing its effectiveness in enhancing model generalization.

**How dose the Prediction Reward (PR) Facilitate Reasonable Assessment?** We further study the importance of the proposed prediction reward. Table 2 shows that adding the prediction reward effectively improves the preference profile prediction accuracy. Besides, Figure 5 shows that incorporating the prediction reward enables accurate profile prediction, which facilitates more reasonable assessment and correct answer, further demonstrating the effectiveness of the prediction reward.

**Personalized Generation with Predicted Preference Profile.** We further discuss how to utilize the predicated profile to generate personalized images in Fig. 6. We first predict a user's preference profile based on reference images. Then, we optimize the initial texts based on the predicated profile

to generate personalized images. The results show that the generated images align well with reference images, affirming the potential of the PreferThinker for facilitating personalized image generation.

**Performance and Speed of PreferThinker without Explicit CoT.** We study whether PreferThinker can implicitly use reasoning capabilities for direct preference assessment. Table 3 shows that it still outperforms SOTA methods on PreferImg and achieves comparable accuracy on PickaPic even without explicit CoT, while delivering the fastest inference speed, further demonstrating its superiority.

**Robustness to Number of Reference Images.** To evaluate the robustness of PreferThinker, we use different number of reference images as user's prior information. Figure 7 shows that PreferThinker outperforms other baselines across different numbers of reference images on both seen and unseen data, revealing its robustness in leveraging limited information for personalized preference assessment.

## 5 CONCLUSION

We introduce PreferThinker, the first reasoning-based personalized image preference assessment system with preference profile prediction. We propose a preference profile to bridge various uses, allowing large-scale user data to be leveraged for training profile prediction and capturing complex personalized preferences. We construct PreferImg-CoT, a CoT-style dataset annotated with preference profiles and high-quality reasoning for interpretability supervision. We adopt a two-stage training strategy comparing Cold-start SFT and reinforcement learning to empower the model with reasoning capabilities. We propose a similarity-aware prediction reward to improve profile prediction, facilitating reasonable assessments. Extensive experiments verify the superiority of PreferThinker.

## ACKNOWLEDGMENTS

This work was supported by National Key RD Program of China under Grant No. 2022YFA1004100.

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

## APPENDIX

For a better understanding of the main paper, we provide additional details in this supplementary material, which is organized as follows:

(**Section A**) We provide more details about the experimental settings.

- We provide more statistics details of the PreferImg test benchmark in Sec. A.1.
- We provide more training details for the SFT cold-start in Sec. A.2.
- We provide more training details for the reinforcement learning post-training in Sec. A.3.

(**Section B**) We provide more details about the constructed dataset PreferImg.

- We provide more details of visual elements comprising the preference profile in Sec. B.1.
- We show several examples of the proposed PreferImg in Sec. B.2.

(**Section C**) We provide additional details about our designed prompt templates.

- We provide prompt templates for generating CoT-style cold-start data with Claude 3.7 in Sec. C.1.

Table S1: PreferImg test benchmark data categories and statistics

| Data Categories | | Number |
|---|---|---|
| In-Domain | Singe-preference | 500 |
| | Multi-preferences | 250 |
| Out-of-Domain | Singe-preference | 500 |
| | Multi-preferences | 250 |
| Total | | 1500 |

- We provide prompt templates using during SFT cold-start and RL post-training phase in Sec. C.2.

(**Section D**) We present more visualization results of personalized image preference assessment on the proposed dataset PreferImg and real user dataset PickaPic.

- We show more visualization results on the PreferImg where users's reference images reflect single preference in Sec. D.1.

- We show more visualization results on the PreferImg dataset where users's reference images reflect multiple preferences in Sec. D.2.

- We show more visualization results on PickaPic to demonstrate our method's generalizability to real-world users in Sec. D.3.

(**Section E**) We conduct further discussions on the effectiveness and performance of proposed PreferThinker, along with limitation and future work.

- We investigate the effectiveness of the cold-start SFT in Sec. E.1.

- We discuss the transferability of the PreferThinker towards assessment and scoring of single image in Sec. E.2.

- We study the effectiveness of PreferThinker to facilitate personalized image generalization in Sec. E.3.

- We discuss the limitation of PreferThinker and the future work in Sec. E.4.

(**Section F**) We describe how we utilize LLMs to assist our work. (Use of LLMs)

## A    EXPERIMENTAL SETTINGS

### A.1    PREFERIMG TEST BENCHMARK.

To comprehensively evaluate the performance of personalized image preference assessment, we construct a test benchmark comprising 1500 user samples based on the proposed PreferImg dataset. We categorize the samples into single-preference (SP) and multi-preference (MP) groups based on whether reference images reflect singular or multiple preferences. To evaluate the generalization of PreferThinker, we further divide the benchmark into in-domain (ID) and out-of-domain (OD) subsets based on whether users' profiles are included in the training data. The detailed statistics of the benchmark are provided in Table S1.

### A.2    TRAINING DETAILS OF SFT COLD-START PHASE.

We utilize Qwen2.5-VL-7B Instruct model as our base model and perform supervised fine-tuning using LLaMA-Factory (Zheng et al., 2024). We train the model on the proposed CoT-style cold-start dataset PreferImg-CoT for one epoch. The dataset includes 50,000 user samples with a single preference and 10,000 user samples with multiple preferences. we freeze the parameters of the vision tower and the multi-modal projector, and fine-tune the LLM. We employ DeepSpeed Zero-3 optimization strategy to handle the memory requirements of large models. The training settings of SFT are detailed in the Table S2.

Table S2: Training settings for cold-start SFT stage.

| batch size | 1 | maximum gradient norm | 1 | precision | bf16 |
|---|---|---|---|---|---|
| gradient accumulation | 2 | learning rate scheduler | cosine | epochs | 1 |
| learning rate | 1e-6 | max length | 32768 | times | 21.1h |
| optimizer | AdamW | deepspeed | zero2 | GPU | 8x A800 |
| warm up ratio | 0.1 | weight decay | 0.0 | trainable module | LLM |

Table S3: Training settings for GRPO-based RL training stage.

| batch size per device | 2 | num of rollout | 6 | precision | bf16 |
|---|---|---|---|---|---|
| gradient accumulation | 2 | $\beta$ | 0.04 | epochs | 0.5 |
| learning rate | 1e-6 | temperature | 0.9 | times | 73h |
| optimizer | AdamW | deepspeed | zero3 | GPU | 6x A100 |
| warm up ratio | 0.03 | weight decay | 0.01 | trainable module | LLM |

### A.3 TRAINING DETAILS OF REINFORCEMENT LEARNING POST-TRAINING PHASE.

Following the cold-start SFT, we employ GRPO-based reinforcement learning for post-training. We utilize the VLM-R1 framework for this phase. For GRPO training, we use 10,000 user samples with single preference and another 10,000 user samples with multiple preferences. The number of reference frames pairs (preferred and non-preferred images) used in training is 5. The training data of the phase does not include CoT-style reasoning, since we aim to incentivize the model to explore more reasonable assessment paths. The training settings of RL are detailed in the Table S3

## B DATASET DETAILS

### B.1 VISUAL PREFERENCE ELEMENTS OF PREFERENCE PROFILE.

To formalize a user's complex personalized visual preferences, we identify 15 visual elements that most frequently appear in Lexica's [2] text prompts and strongly influence user preference toward personalized image generation. We then conduct a user study with 100 participants, asking each to select the five most important visual elements as visual preference profile. The result in Fig S1(a) reveals that *art style*, *color*, *detail*, *art medium* and *saturation* are voted as the most representative visual elements for characterizing personalized preferences. To ensure preference profile diversity, we collect a rich vocabulary of related terms for each visual element, as shown in Fig. S1(b), laying a solid foundation for constructing a large-scale and diverse dataset for personalized image preference assessment.

To ensure the diversity of personalized visual preference profiles, we collect an extensive set of vocabulary related to five key visual elements, as illustrated in Fig. S2. Leveraging this lexicon, we construct a large-scale and rich set of visual preference profiles, laying a robust foundation for constructing the dataset PreferImg and CoT-style dataset PreferImg-CoT.

### B.2 EXAMPLES OF THE PROPOSED DATASET PREFERIMG.

We visualize several user examples of the proposed dataset PreferImg in Fig. S3 and Fig. S4. Figure. S3 shows examples where users' reference images reflect single preference, while Fig. S4 presents examples where users' reference images reflect multiple preferences. It is observed that our dataset encompasses a wide variety of image content and diverse visual preference profiles.

---

[2]https://lexica.art/

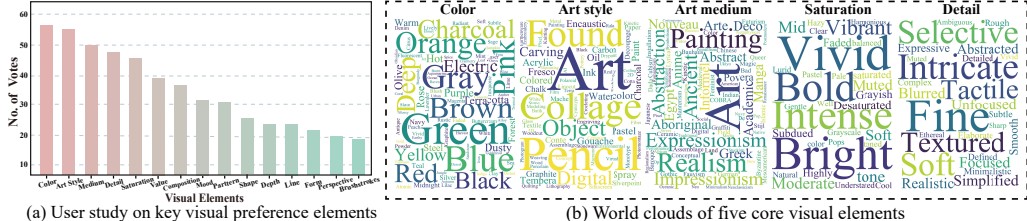

| (a) User study on key visual preference elements | (b) World clouds of five core visual elements |

Figure S1: Key visual elements of preference profile. (a) User study result reveals that color, art style, art medium, saturation, and detail are voted five key elements representing the visual preference profile. (b) World clouds show that each element has a rich vocabulary associated with it.

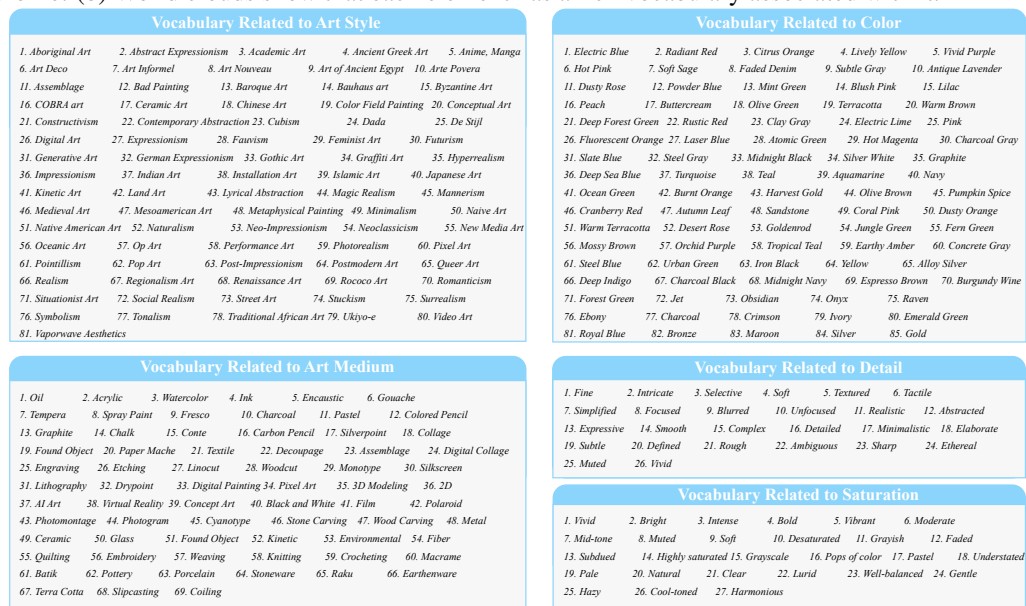

Figure S2: Vocabulary related to five core visual preference elements.

## C PROMPT TEMPLATES

### C.1 PROMPT TEMPLATES FOR COT-STYLE COLD-START DATA GENERATION.

To enable the Claude 3.7 model to annotate high-quality, CoT-style assessments for the PreferImg dataset, we design a detailed prompting strategy that guides the Claude 3.7. This strategy directs Claude to use the meta information, including (1) user visual preference and non-preference profile, (2) question prompt, and (3) assessment answer to provide an explainable score and assessment for candidate images. The prompt mainly consists of an evaluation template and guidelines, follows a "predict-then-assess" structure. First, based on a reference image, the model predicts the user's preference profile. Then, using this predicted profile, it evaluates the candidate images across multiple dimensions and assigns interpretable scores. Finally, the model selects the user's preferred image based on the total scores. The prompts for generating the CoT-style assessment data are shown in Fig. S5 and Fig. S6. Figure S5 shows the prompt for single-preference users, while Figure S6 shows the prompt for multi-preference users.

### C.2 PROMPT TEMPLATES USING DURING SFT AND RL TRAINING.

We provide prompt templates for use during cold-start Supervised Fine-Tuning (SFT) and Group Policy Reinforcement Learning (GRPO RL) training. These templates, which include a system prompt and a question prompt, describe the specific task and provide correct output examples. This guides the model to perform explainable, personalized preference assessments for candidate images in the correct format. The prompt templates for SFT and RL training are shown in Fig. S7 and Fig. S8. Figure S7 illustrates the prompt for single-preference users, while Figure S8 shows the prompt for multi-preference users.

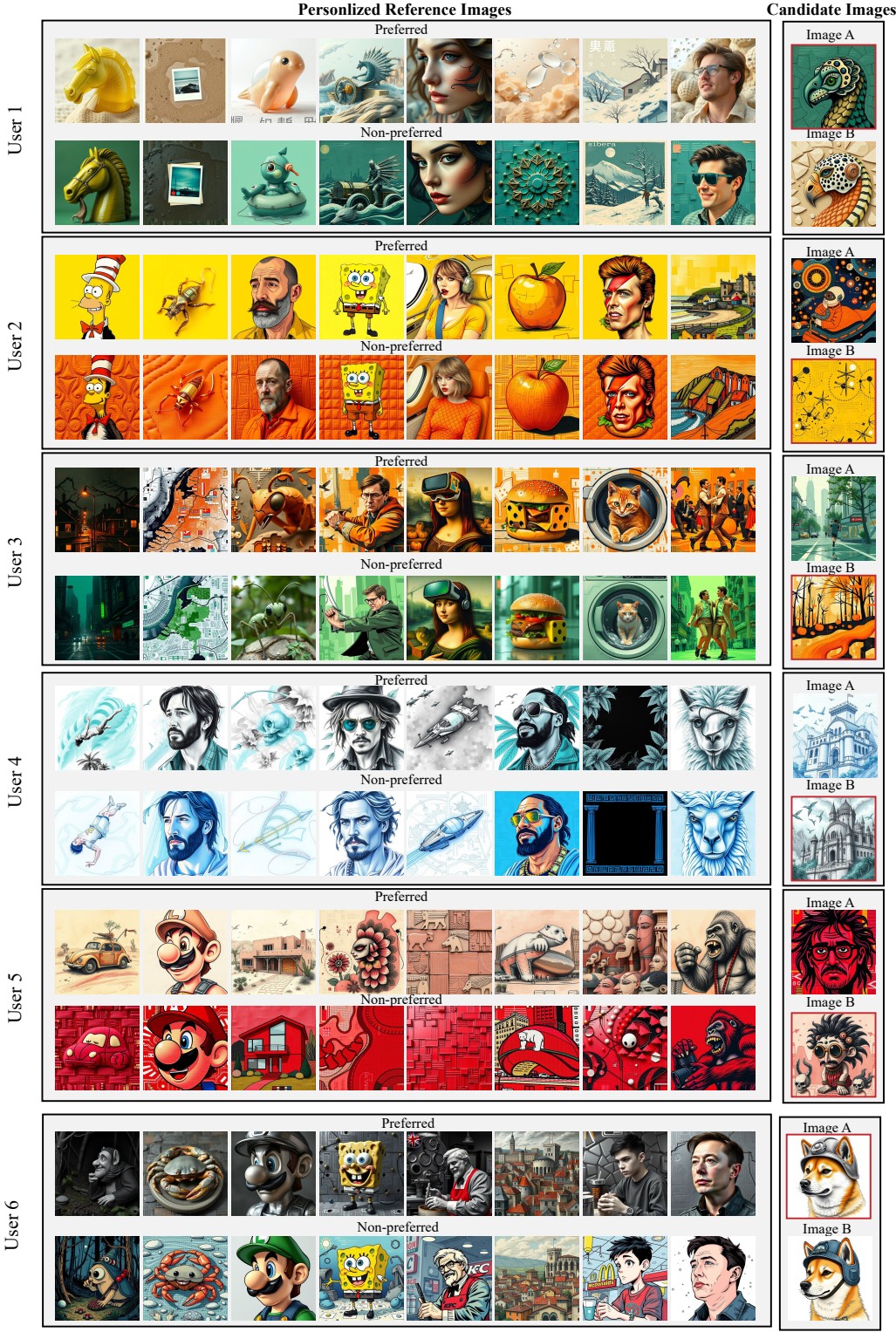

Figure S3: Examples of the proposed dataset PreferImg, where users' reference images reflect single preference. Red box indicates preferred image in the candidate images.

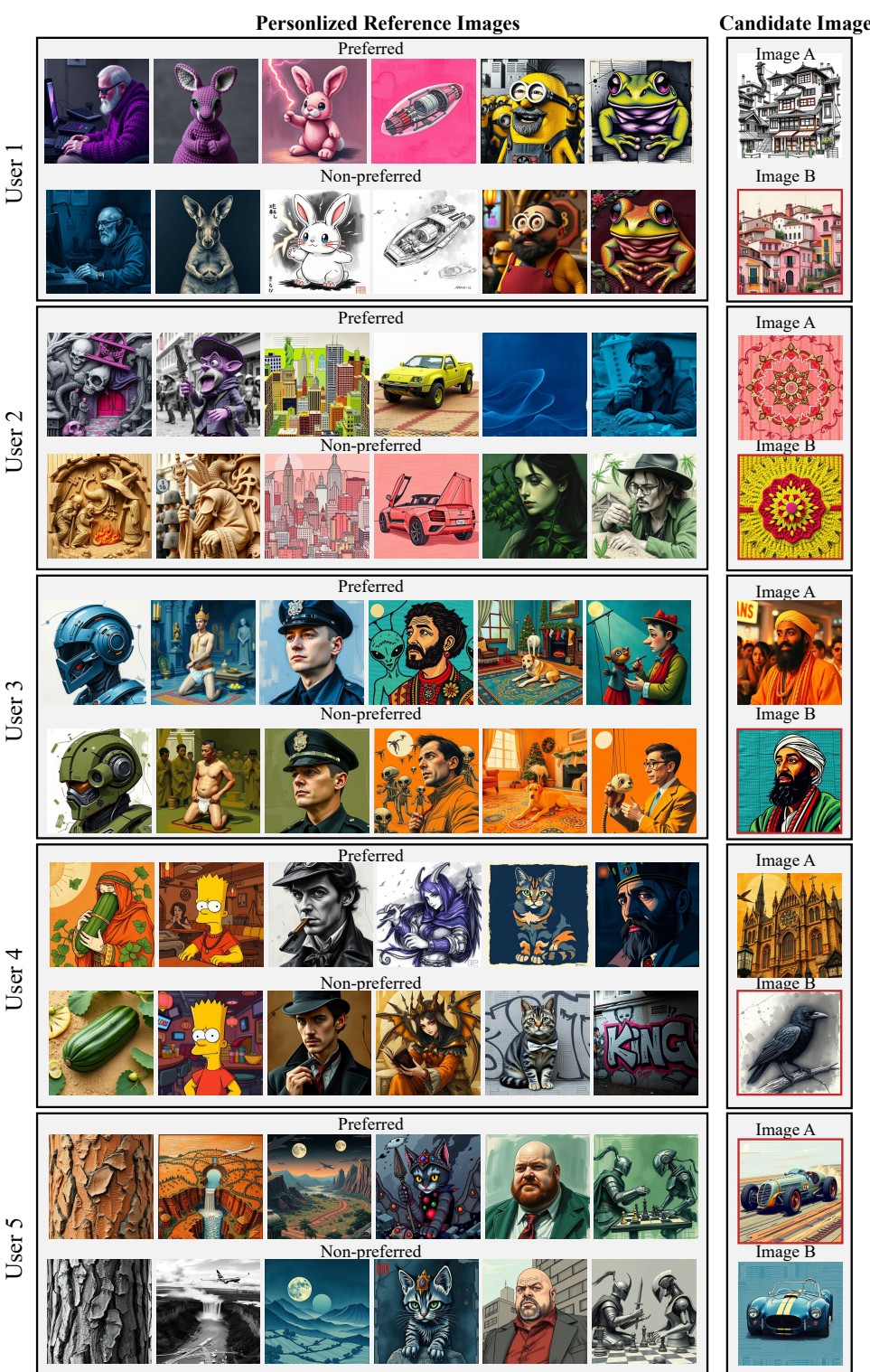

Figure S4: Examples of the proposed dataset PreferImg, where users' reference images reflect multiple preferences. Red box indicates preferred image in the candidate images.

# D VISUALIZATION RESULTS

## D.1 VISUALIZATION RESULTS ON USERS WITH SINGLE PREFERENCE

We present more visualization results on the PreferImg where users's reference images reflect single preference in Figures S9 to S11. It is observed that PreferThinker first predicts the users' preference

and non-preference profiles from the reference images, and then provides a reliable assessment and score for candidate images based on these profiles, which not only enables accurate assessment but also provides interpretability.

### D.2 VISUALIZATION RESULTS ON USERS WITH MULTIPLE PREFERENCES

We show more visualization results on the PreferImg where users's reference images reflect multiple Preferences in Fig. S12 and S13. Our model can accurately predict multiple preference profiles from reference images and provide a reliable assessment and score for candidate images based on these profiles, revealing the effectiveness of PreferThinker for users with multiple preferences.

### D.3 VISUALIZATION RESULTS ON REAL USERS

To demonstrate the generalizability of our method to real-world users, we further utilize the Prefer-Thinker to conduct personalized preference assessment for real users from PickaPic. As shown in Figures S14 to S17 Although the preferences in real users' reference images are more complex, our method can still effectively extract their primary preference profiles and provide a correct assessment.

## E DISCUSSION

### E.1 IMPORTANCE OF COLD START INITIALIZATION.

We further discuss the importance of cold-start SFT for correct structured reasoning. Figure S18(a) shows that directly using the base model (Qwen-2.5-VL) fails to provide numerical scores and reasonable reasoning and leads to incorrect answer. Figure S18 (b) shows that while applying reinforcement learning alone on the based model can generate the total numerical scores, it fails to provide the correct output format and interpretable scores for each visual elements. In contrast, by first teaching the model proper structural reasoning with Supervised Fine-Tuning (SFT) before RL, the model can provide reasonable, explainable scores for each dimension, ultimately leading to a correct final assessment based on the total score, as shown in Figure S18 (c).

### E.2 TRANSFERABILITY TO PERSONALIZED ASSESSMENT AND SCORING OF SINGLE IMAGE.

We further study the transferability of PreferThinker towards personalized assessment and scoring of single image. We directly modify the question prompt to enable the PreferThinker to provide an assessment and score for a single generated image. The visualization results are shown in Fig. S19 and Fig. S20. We can observe that PreferThinker can flexibly provide an interpretable assessment and score for a single image.

### E.3 PERSONALIZED IMAGE GENERATION WITH PREDICTED PROFILE.

To further demonstrate the benefits of the proposed preference profile, we employ the predicted preference profile to achieve personalized image generation in Fig. S21. We first predict the user's preference profile based on reference images. Then, we optimize the initial text based on the profile to generate personalized images.The results show that the personalized images align well with reference images, revealing the potential of the proposed method to facilitate personalized generation.

### E.4 LIMITATION AND FUTURE WORK

The main limitation of our method is that the personalized preference dataset built on synthetic profiles may not fully capture the complex and subtle real human preferences. Our current preference profile only contains five key visual attributes that influence human visual preferences. However, real users' preferences are often complex and subtle, shaped by multiple factors. Beyond visual attributes, they are also closely related to image content semantics and users' backgrounds (e.g., age, gender and culture). Consequently, our method may struggle with complex and hard-to-articulate real user preferences, as shown in Fig. S22. In future work, we will incorporate additional key factors into preference profiles to more comprehensively capture real users' preferences.

---

**Prompt for Claude to Generate CoT-style Data (User with Single Preference)**

Given a user's historical preferred reference images, non-preferred reference images, and two candidate images, along with the following meta information: 1. User's visual preference and non-preference profile; 2.Question prompt; 3. Assessment Answer.

You are an personlized preference analysis assistant. Please strictly follow the requirements below to analyze the candidate images and generate Chain-of-Thought style output:

• Your final output must include the following four tags:
<visual preference profile>, <visual non-preference profile>, <think>, and <answer>

• Within the <visual preference profile> tag, directly copy the visual preference profile provided in the user meta information in JSON key-value format, for example:
**<visual preference profile>**
Art style: Manga; Color: Blush Pink; Art Medium: Digital Painting; Saturation: Moderate Detail: Fine
**</visual preference profile>**

• Within the <visual non-preference profile> tag, directly copy the visual non-preference profile provided in the user meta information in JSON key-value format, for example:
**<visual non-preference profile >**
Art style: Graffiti Art; Color: Powder Blue; Art Medium: Acrylic Painting; Saturation: Vibrant; Detail: Soft
**</visual non-preference profile>**

• Within the <think> tag, based on the preference and non-preference profile, score and reason about Image A and Image B across five dimensions: "Art style", "Color", "Art Medium", "Saturation", and "Detail". The scoring criteria are as follows:
1.  Score principle:
Preference attributes: If image features highly match preference attributes (e.g., rich details), assign a high score (4-5).
Non-preference attributes: If image features resemble non-preference attributes (e.g.lack of detail), assign a low score (1-2).
Neutral features: Assign a medium score (3) when there is no significant tendency.2
2.  Naturally incorporate keywords in the reason analysis:
Attribute terms (e.g., Performance Art) must be integrated into descriptions of specific scene elements. Listing them alone or in quotation marks is prohibited.
e.g."The dynamic street graffiti uses spontaneously splattered paint, and its fluidity highly corresponds to the live-creation characteristic of Performance Art."
Output format must strictly follow:
**<think>**
1. Artistic style:
Image A({score}) - {reason};  Image B({score}) - {reason}
2. Color:
Image A({score}) - {reason};  Image B({score}) - {reason}
......
Total:
Image A: total_score={score1}+{score2}+{score3}+{score4}+{score5}={total_score_A}
Image B: total_score={score1}+{score2}+{score3}+{score4}+{score5}={total_score_B}
**</think>**

• Within the <answer> tag, directly output the final choice provided in the user prompt ("Image A" or "Image B"), for example:
**<answer>**Image A**</answer>**

Figure S5: Prompt for Claude 3.7 to generate CoT-style assessment for users with single preference.

# F USE OF LLMS

We only used LLMs to polish and correct the text and grammar in the Abstract and Introduction sections.

**Prompt for Claude to Generate CoT-style Data (User with Multiple Preferences)**

Given a user's historical preferred reference images, non-preferred reference images, and two candidate images, along with the following meta information: 1. User's **multiple** visual preference and non-preference profile; 2.Question prompt; 3. Assessment Answer.

You are an personlized preference analysis assistant. Please strictly follow the requirements below to analyze the candidate images and generate Chain-of-Thought style output:

• Your final output must include the following four tags:
<visual preference profile>, <visual non-preference profile>, <think>, and <answer>

• Within the <visual preference profile> tag, directly copy the visual preference profile provided in the user meta information in JSON key-value format. Please note that users may have multiple preference profiles. Here is an example:
**<visual preference profile>**
**1. Preference profile 1**
Art style: Manga; Color: Blush Pink; Art Medium: Digital Painting; Saturation: Moderate Detail: Fine
**2. Preference profile 2**
Art style: Ceramic Art; Color: Vivid Purple; Art Medium: Acrylic; Saturation: Bright Detail: Focused
......
**</visual preference profile>**

• Within the <visual non-preference profile> tag, directly copy the visual non-preference profile provided in the user meta information in JSON key-value format. Please note that users may have multiple preference profiles. Here is an example:
**<visual non-preference profile >**
**1. Non-preference profile 1**
Art style: Graffiti Art; Color: Powder Blue; Art Medium: Acrylic Painting; Saturation: Vibrant; Detail: Soft
**2. Non-preference profile 2**
Art style: Ukiyo-e; Color: Hot Pink; Art Medium: Spray Paint; Saturation: Grayish; Detail: Blurred
......
**</visual non-preference profile>**

• Within the <think> tag, based on the preference and non-preference profile, score and reason about Image A and Image B across five dimensions: "Art style", "Color", "Art Medium", "Saturation", and "Detail". The scoring criteria are as follows:
1. Score principle:
Preference attributes: If image features highly match preference attributes (e.g., rich details), assign a high score (4-5).
Non-preference attributes: If image features resemble non-preference attributes (e.g.lack of detail), assign a low score (1-2).
Neutral features: Assign a medium score (3) when there is no significant tendency.2
2. Naturally incorporate keywords in the reason analysis:
Attribute terms (e.g., Performance Art) must be integrated into descriptions of specific scene elements. Listing them alone or in quotation marks is prohibited.
e.g."The dynamic street graffiti uses spontaneously splattered paint, and its fluidity highly corresponds to the live-creation characteristic of Performance Art."
Output format must strictly follow:
**<think>**
1. Artistic style:
Image A({score}) - {reason};  Image B({score}) - {reason}
2. Color:
Image A({score}) - {reason};  Image B({score}) - {reason}
......
Total:
Image A: total_score={score1}+{score2}+{score3}+{score4}+{score5}={total_score_A}
Image B: total_score={score1}+{score2}+{score3}+{score4}+{score5}={total_score_B}
**</think>**

• Within the <answer> tag, directly output the final choice provided in the user prompt ("Image A" or "Image B"), for example:
**<answer>**Image A**</answer>**

Figure S6: Prompt for Claude 3.7 to generate CoT-style data for users with multiple preferences.

**Prompt Template for SFT and RL Training (User with Single Preference)**

**System Prompt:** You are an personlized preference analysis assistant.The user provides a set of historically preferred and non-preferred images. Based on this historical data, please predict the user's visual preference profile and non-preference profile across five key visual elements: art style, color, artistic medium, saturation, and detail. The predicted results should be described using terms corresponding to each of these five visual attributes.Subsequently, the user provides two candidate images: Image A and Image B. Using the predicted visual preference profile, assign an interpretability score (0–5) for each of the five attributes for both images. A higher score indicates stronger alignment with the user's preference profile, while a lower score suggests greater similarity to non-preference profile. Each score must be accompanied by a rationale explaining the rating.Finally, sum the scores across all five attributes for each image. The image with the higher total score should be identified as the one the user is more likely to prefer.
Below are some examples:
{Reasoning-based output example 1}
{Reasoning-based output example 2}
......

**Question Prompt:** Given a user's reference images (both preferred and non-preferred images) and two candidate images, first predict the user's visual preference and non-preference profile based on the reference images, then assess which of the two candidate images better matches the user's personlized preference.

Figure S7: The SFT and RL prompt template for users with single preference

> **Prompt Template for SFT and RL Training (User with Multiples Preferences)**
>
> **System Prompt:** You are an personlized preference analysis assistant. The user provides a set of historically preferred and non-preferred images. Based on this historical data, please predict the user's visual preference profile and non-preference profile across five key visual elements: art style, color, artistic medium, saturation, and detail. **Please note that user may has multiple preference profiles.** The predicted results should be described using terms corresponding to each of these five visual attributes. Subsequently, the user provides two candidate images: Image A and Image B. Using the predicted visual preference profile, assign an interpretability score (0–5) for each of the five attributes for both images. A higher score indicates stronger alignment with the user's preference profile, while a lower score suggests greater similarity to non-preference profile. Each score must be accompanied by a rationale explaining the rating. Finally, sum the scores across all five attributes for each image. The image with the higher total score should be identified as the one the user is more likely to prefer.
>
> Below are some examples:
> {Reasoning-based output example 1}
> {Reasoning-based output example 2}
> ......
>
> **Question Prompt:** Given a user's reference images (both preferred and non-preferred images) and two candidate images, first predict the user's visual preference and non-preference profile based on the reference images, then assess which of the two candidate images better matches the user's personlized preference.

Figure S8: The SFT and RL prompt template for users with multiple preferences

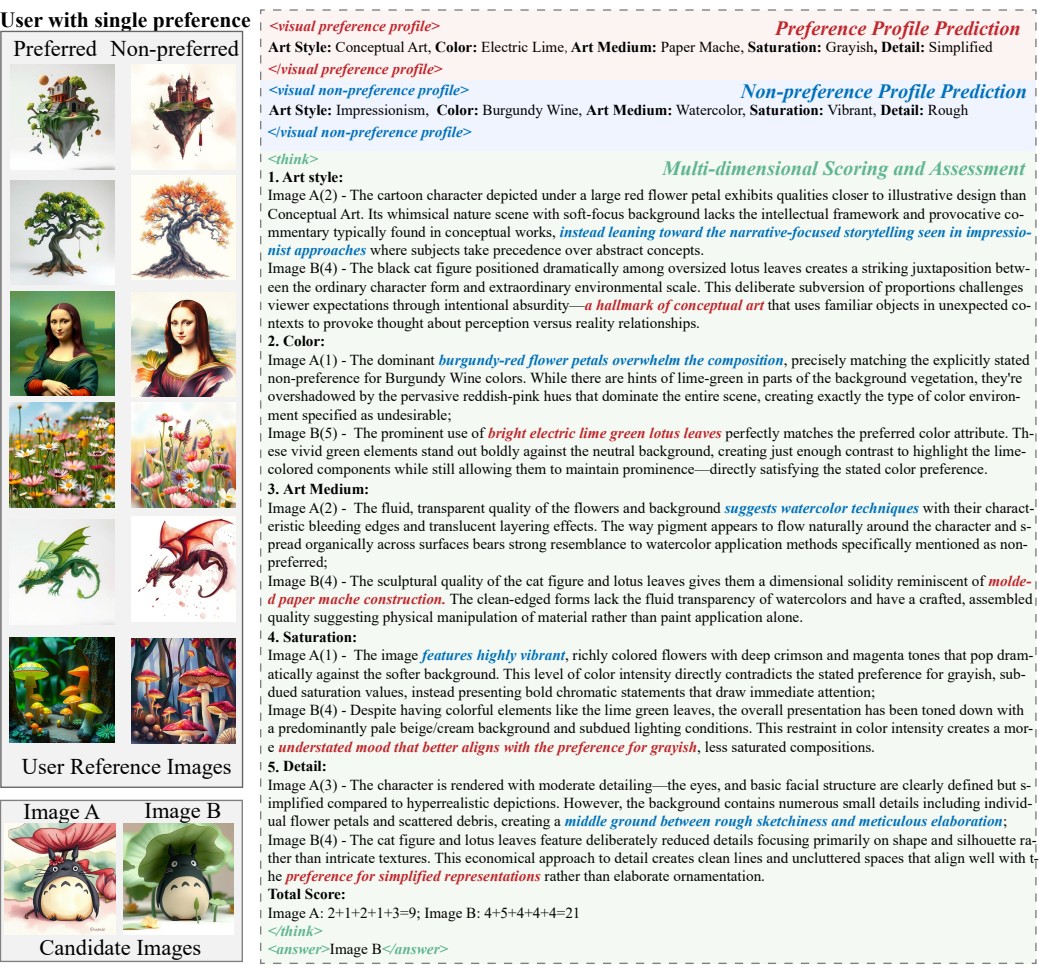

Figure S9: Reasoning-based personalized assessment for users with single preference.

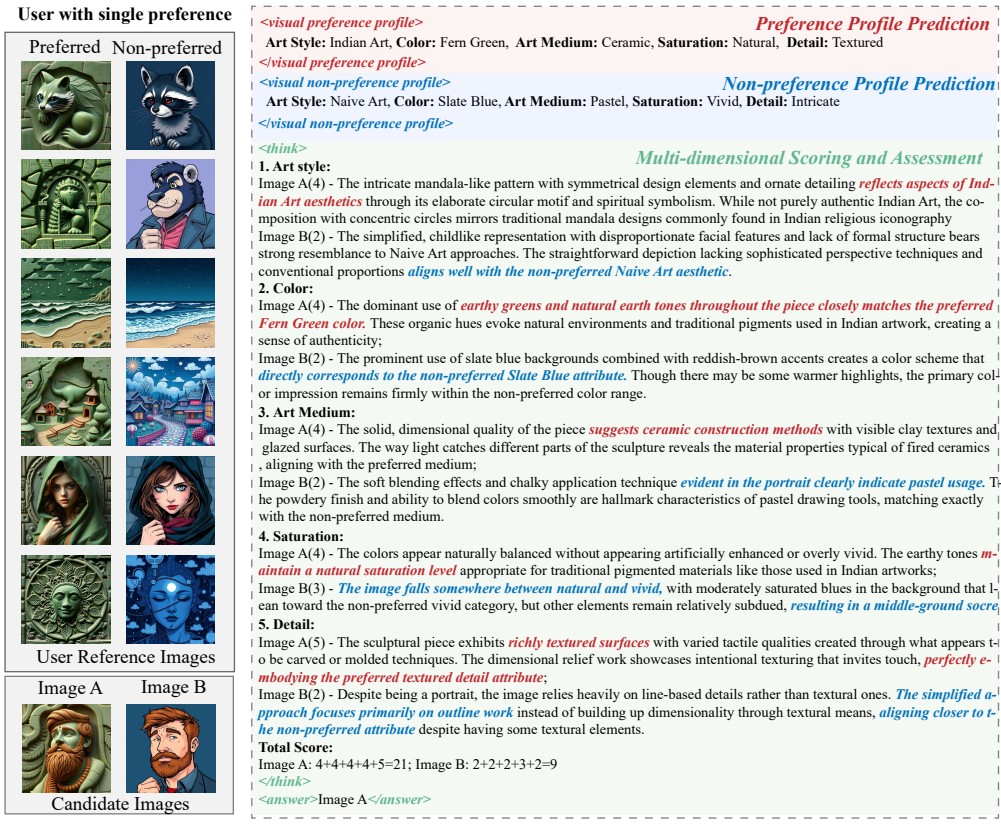

Figure S10: Reasoning-based personalized assessment for users with single preference.

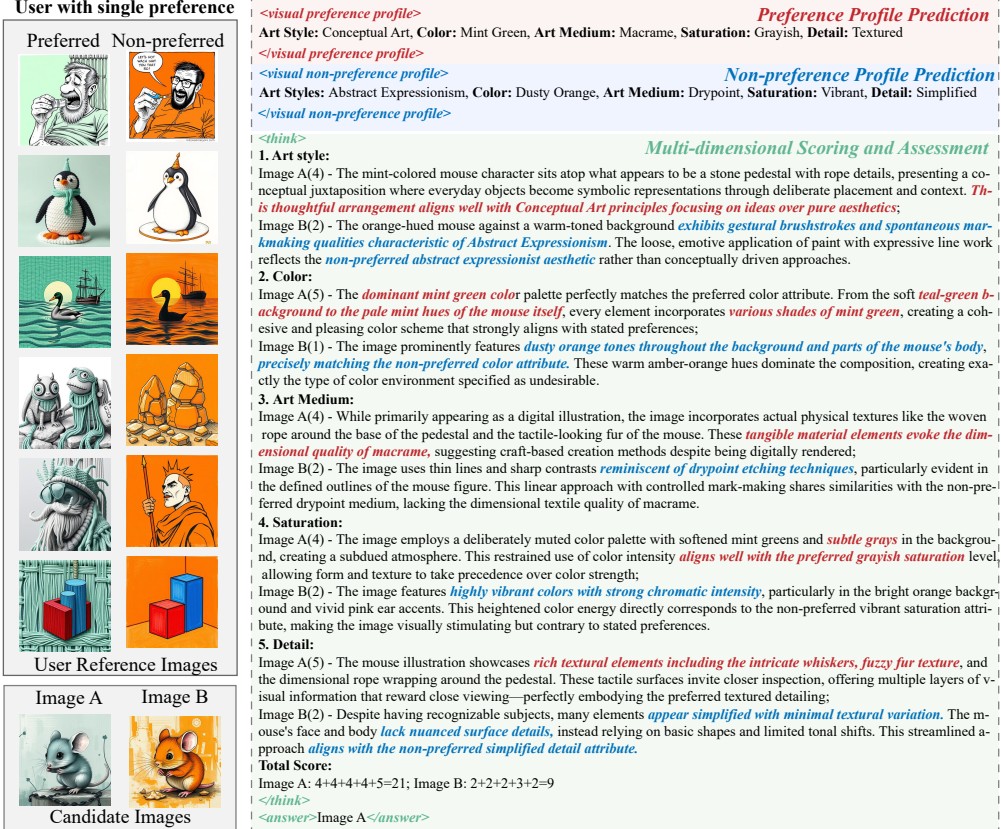

Figure S11: Reasoning-based personalized assessment for users with single preference.

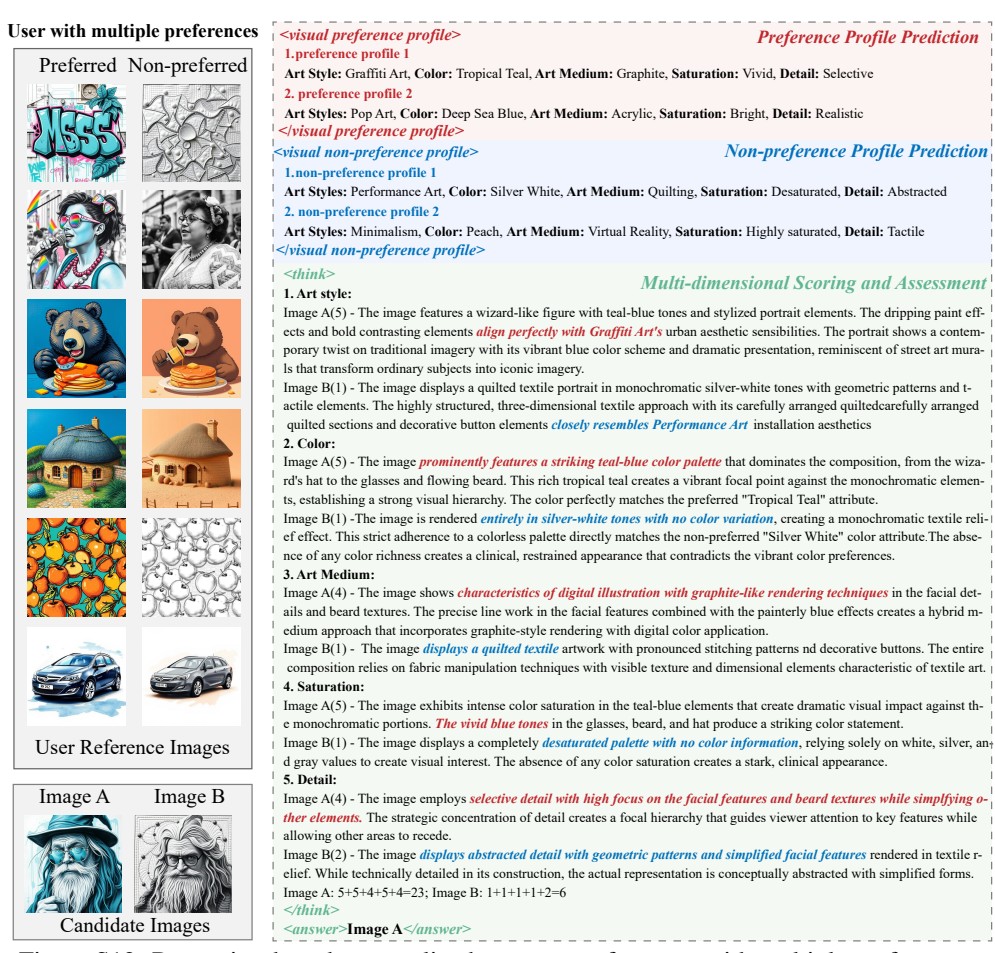

Figure S12: Reasoning-based personalized assessment for users with multiple preferences.

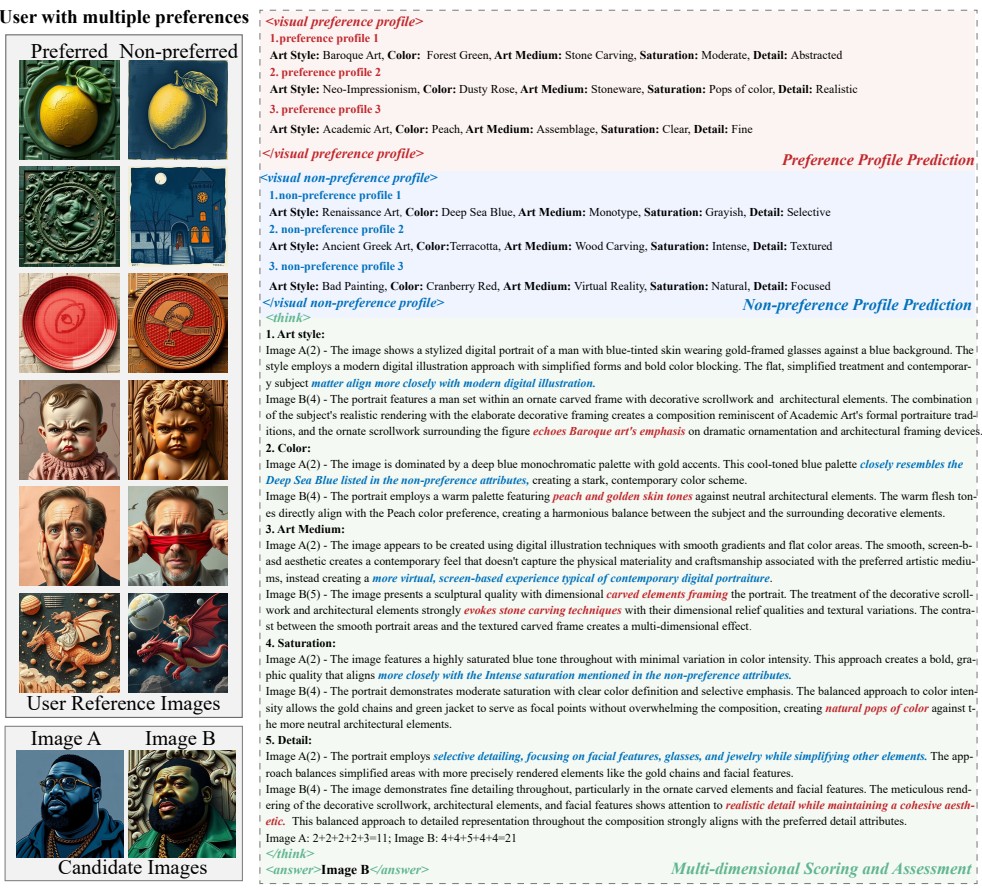

Figure S13: Reasoning-based personalized assessment for users with multiple preferences.

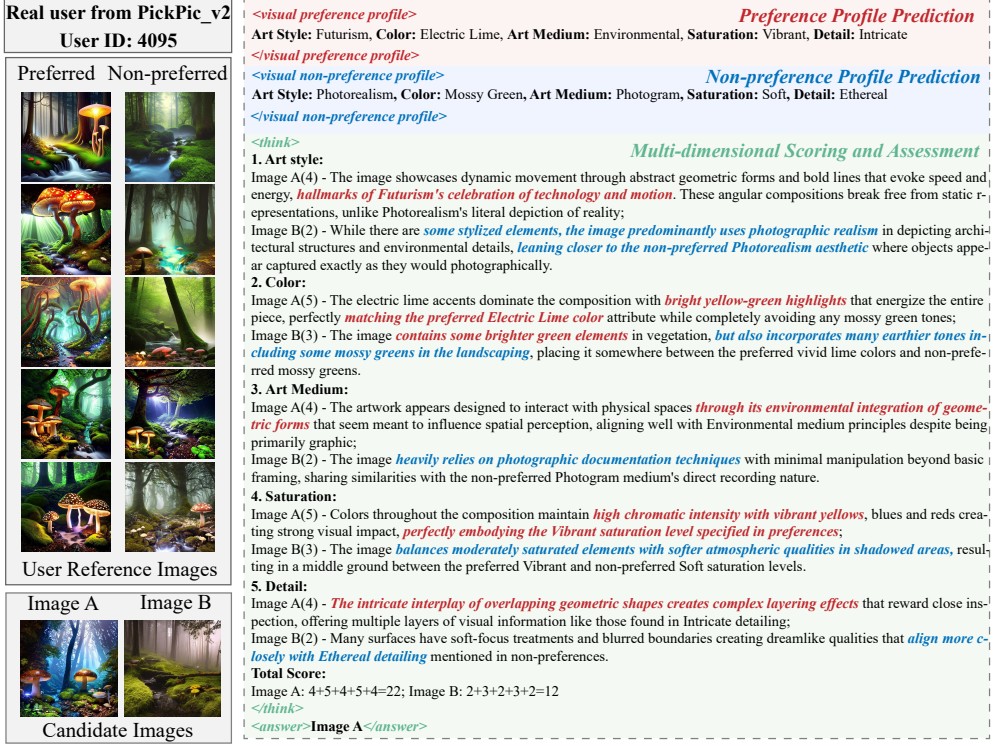

Figure S14: Reasoning-based personalized assessment for Real user from PickaPic.

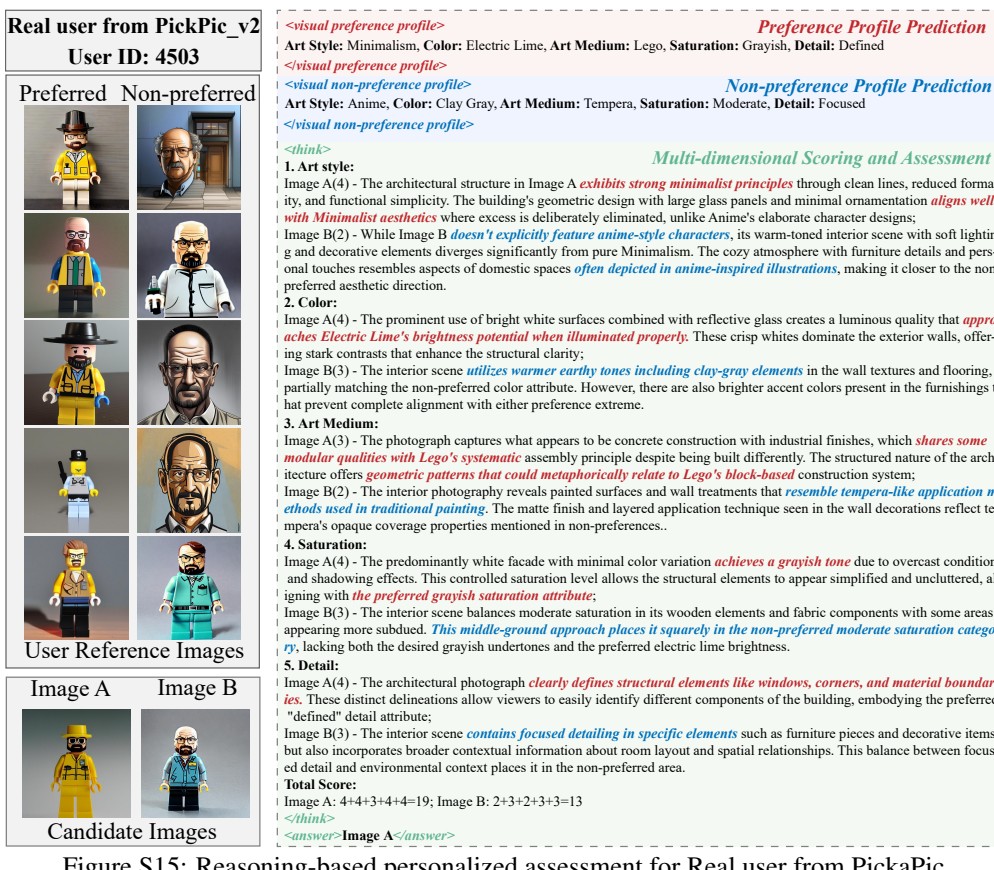

Figure S15: Reasoning-based personalized assessment for Real user from PickaPic.

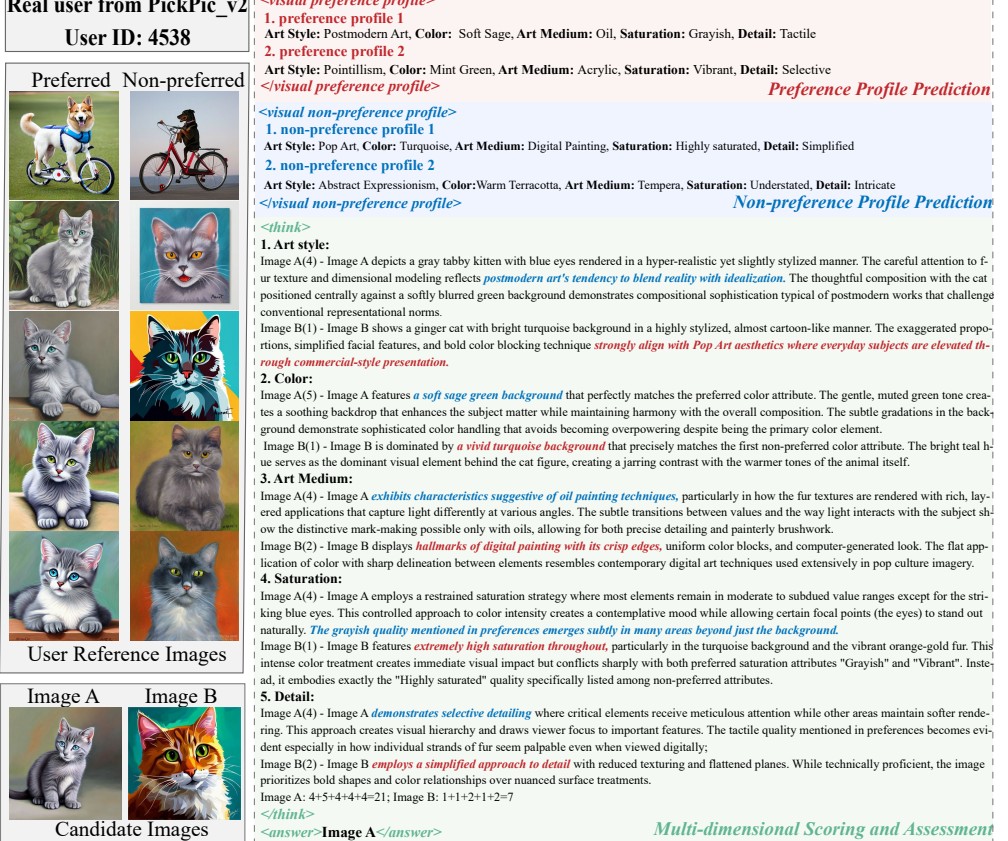

Figure S16: Reasoning-based personalized assessment for Real user from PickaPic.

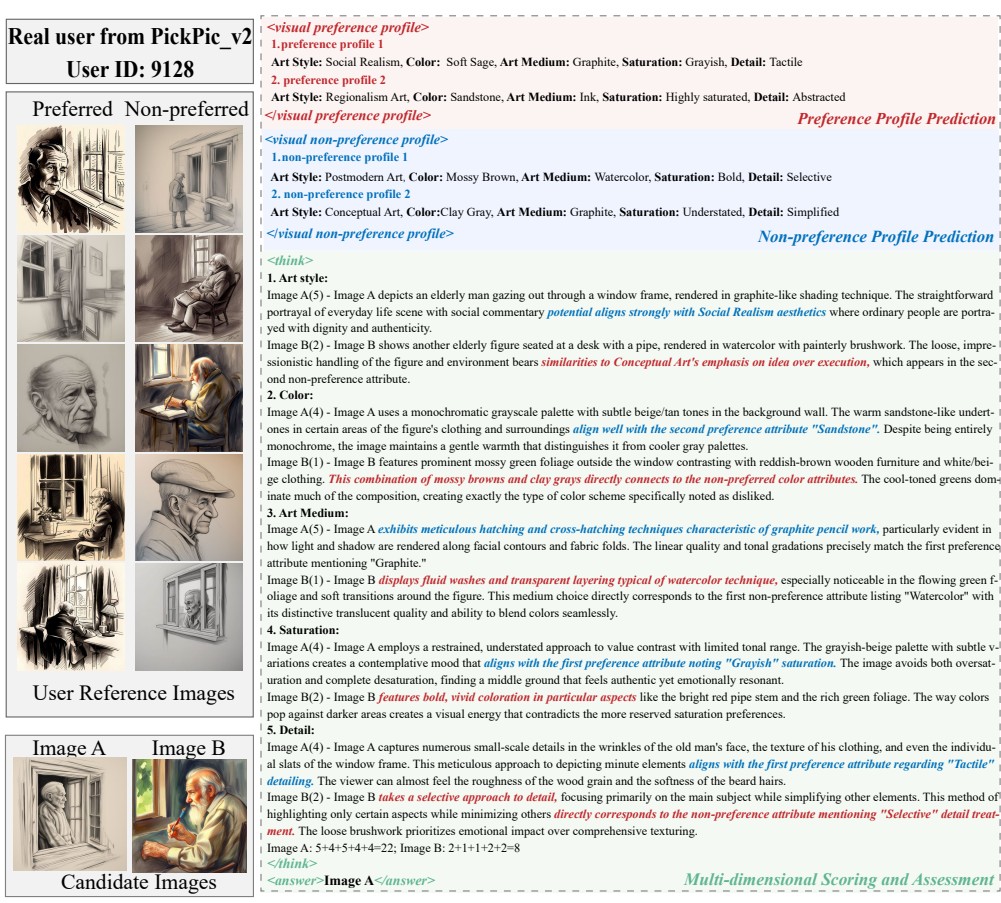

Figure S17: Reasoning-based personalized assessment for Real user from PickaPic.

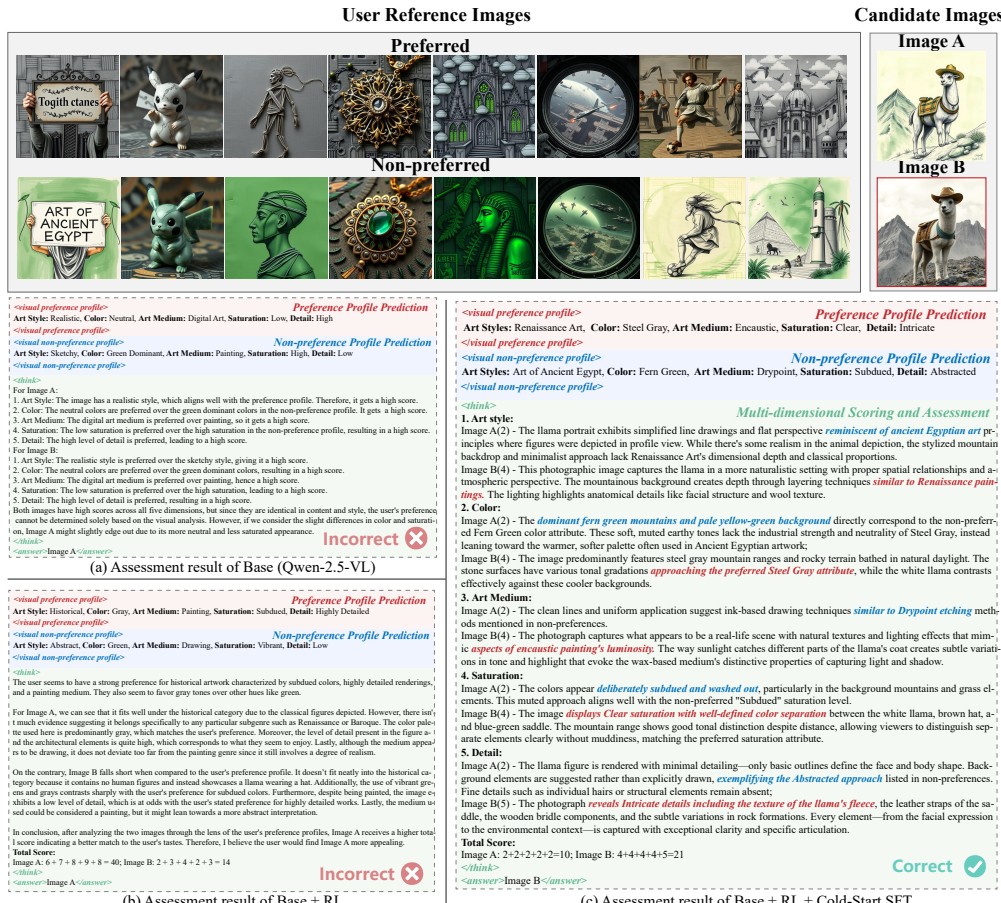

Figure S18: Effectiveness of cold-start SFT for correct and structured reasoning.

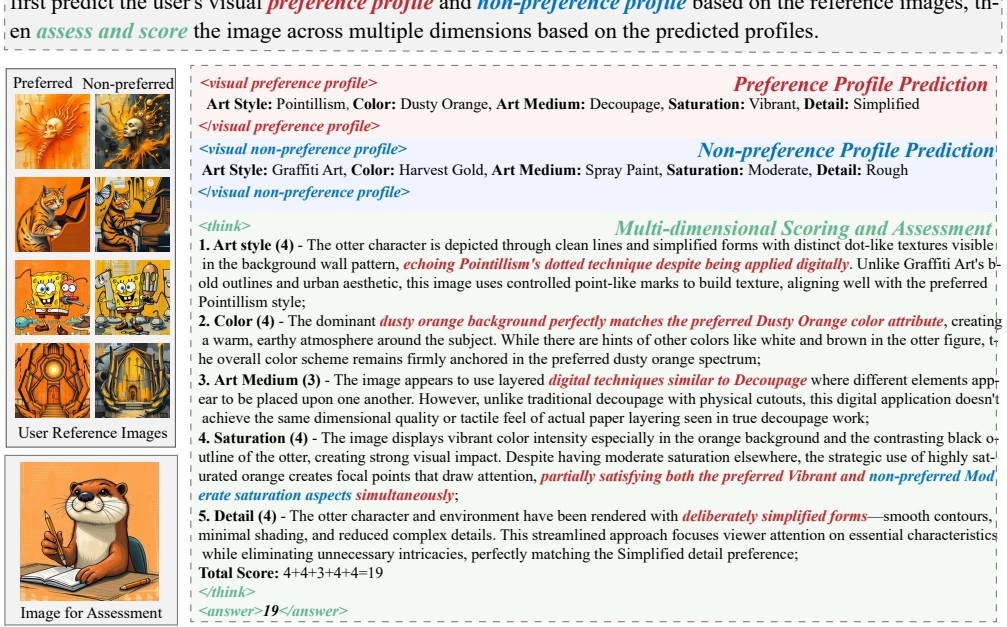

Figure S19: Reasoning-based personalized assessment and scoring for single image.

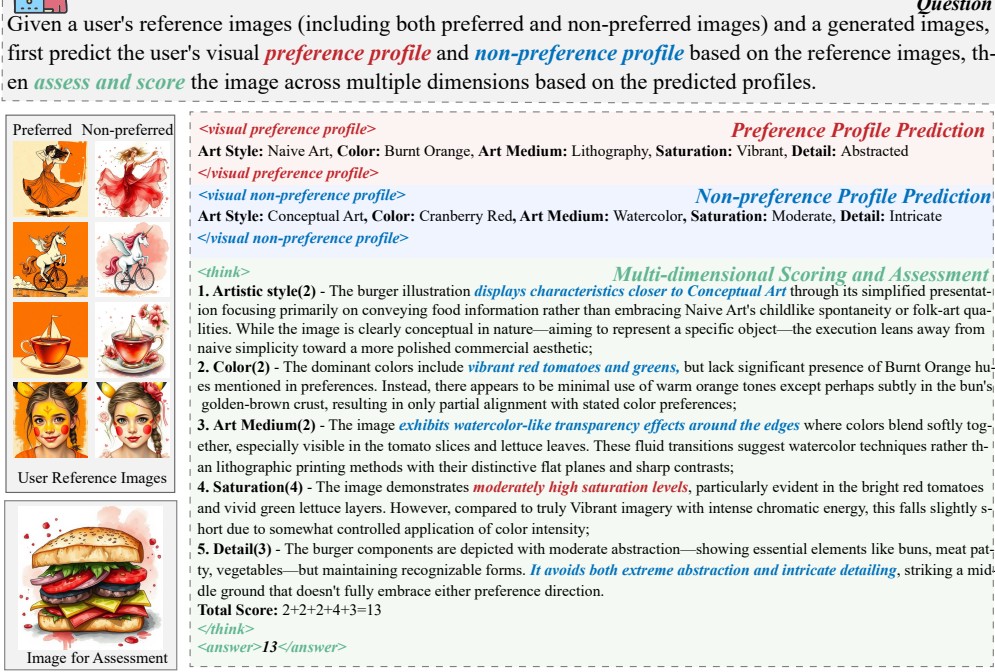

Figure S20: Reasoning-based personalized assessment and scoring for single image.

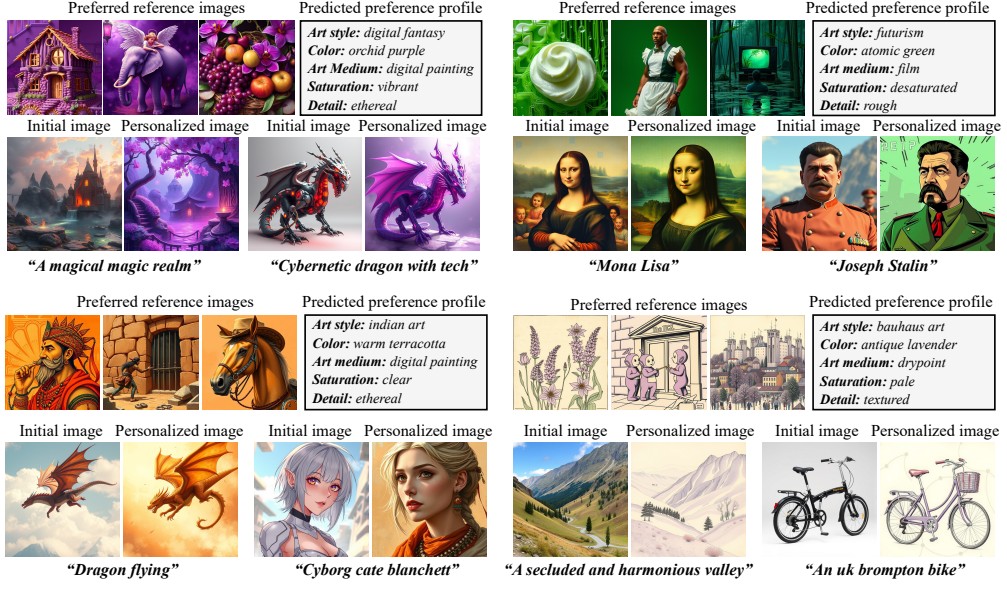

Figure S21: Personalized image generalization with the predicted profile.

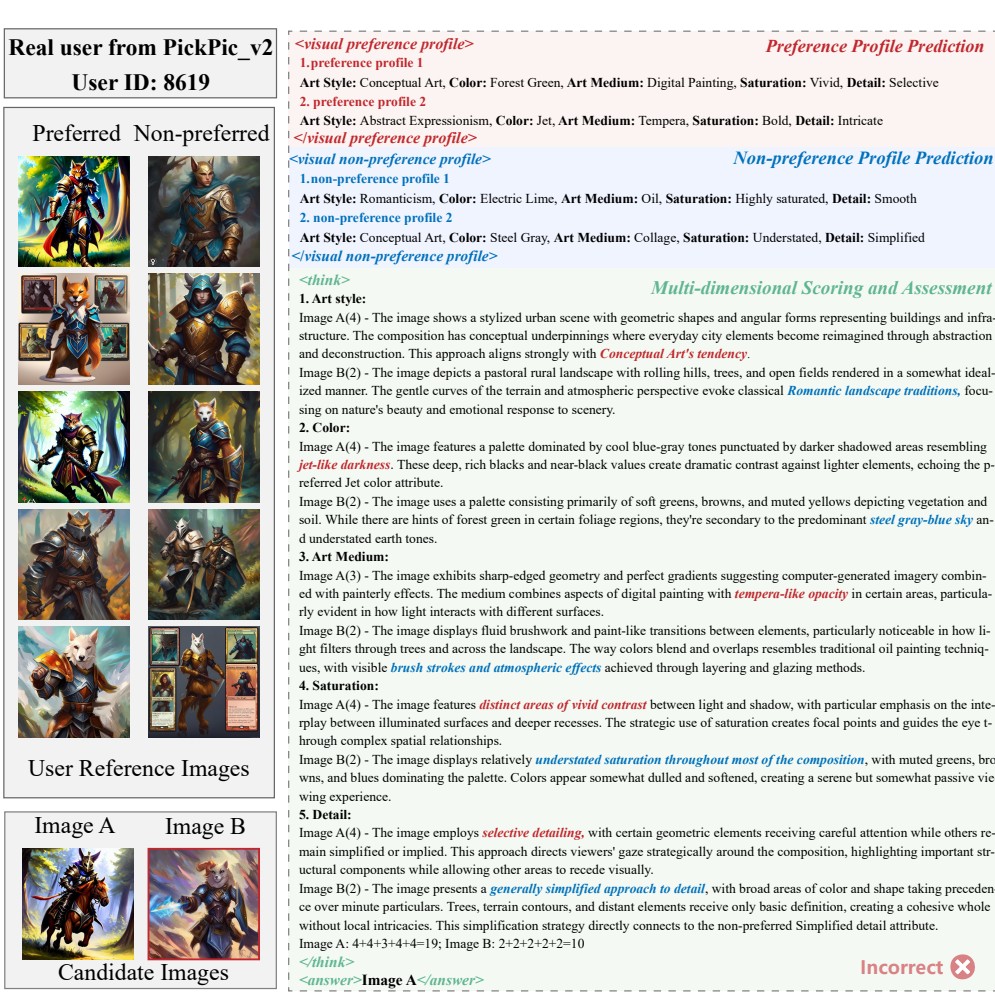

Figure S22: Failed case of the Preferthinker for real users with complex preferences.

