# OpenReview forum: "PreferThinker: Reasoning-based Personalized Image Preference Assessment"
_ICLR.cc/2026/Conference — ICLR 2026 Poster_

### Official Review · Reviewer_oEnc · 2025-10-21

**Soundness:** 3
**Presentation:** 3
**Contribution:** 3
**Rating:** 6
**Confidence:** 3

**Summary:**

This paper proposes a new CoT-based methodology for personalized image preference assessment and introduces a large-scale CoT-style dataset. Most existing work captures general human preferences and cannot handle personalized ones. The paper introduces the concept of "common preference profile" and proposes a two-stage "predict-then-assess" method. The interpretable preference profile and reasoning steps help MLLMs to achieve better alignment for each user's personalized image synthesis.

**Strengths:**

1. The innovative “Common Preference Profile” efficiently bridges the gap between large-scale general preference alignment and personalized preference modeling.
---
2. The proposed "predict-then-assess" paradigm decomposes the task into *profile prediction* (summarizing human preference across several pre-defined dimensions) and *assessment reasoning* (interpretation and scoring). This decomposition provides **better modularity for image generation** and interpretability.
---
3. The paper empirically verifies that carefully designed reasoning structures—incorporating CoT reasoning to (1) predict each user's preference profile and (2) assess candidate images with the uncovered preference profile—achieve state-of-the-art performance on human preference prediction.

**Weaknesses:**

The discrete elementary preference profile may oversimplify the real-world human's nuanced preference, as it reduces complex, context-dependent aesthetic judgments into a fixed set of categorical dimensions that may not capture the subtle variations and contradictions inherent in individual taste.

**Questions:**

Humans have inconsistent preferences across different topics and scenarios. How does the proposed elementary preference profile approach address this complexity? Can you clarify the motivation behind using independent element-wise preference profiles rather than capturing the continuous latent representation for users' preferences?

---

> ### Author Response · Authors · 2025-11-20
>
> Dear Reviewer oEnc, we thank you for your valuable comments. We appreciate your insightful questions and hope our responses could address your concerns.
>
> **[W1]: The discrete elementary preference profile may not capture the subtle variations and contradictions inherent in real individual tastes.**
>
> **[A]:** Thanks for your careful review. We agree that the elementary preference profile simplifies real user preferences and may not fully capture complex individual tastes. Nevertheless, our method achieves the second-best performance on the unseen real dataset Pick-a-Pic, surpassing most existing methods, including the state-of-the-art closed-source MLLM. Notably, the first-place method PickScore is trained on Pick-a-Pic, whereas our method does not use this dataset for training, revealing its generalization to real users. We also agree that this generalization is limited, and the model may still fail when faced with more complex and subtle real user preferences. This limitation is discussed in detail in Section E.4 in Appendix and Figure S22.
>
> To better handle complex real users’ personalized preferences, we propose two potential directions for future work:
>
> **1. Constructing a more comprehensive personalized preference profile.** Our current preference profile only contains five key visual attributes that influence human visual preferences. However, real users’ preferences are often complex and subtle, shaped by multiple factors. Beyond visual attributes, they are also closely related to image content semantics and users’ backgrounds (e.g., age, gender and culture).  In future work, we will incorporate additional key factors into preference profiles to more comprehensively capture real users’ preferences.
>
> **2. Generating pseudo-labels of personalized preference profiles for real users.** We can construct a real personalized dataset with “pseudo” preference profile labels. Specifically, we first collect real users’ personalized data, including user IDs, reference images, candidate images and preference selections. We then leverage the strong image understanding capabilities of current advanced closed-source MLLMs (e.g., GPT-5.1) to extract “pseudo” preference profile labels from the users’ reference images and adaptively identify the key factors that determine user preferences. Based on this dataset, the model trained using it can more flexibly capture subtle real users’ preferences.
>
> **[Q1]: How does the elementary preference profile address users’ inconsistent preferences across different topics and scenarios?**
>
> **[A]:** Thanks for your insightful question. First, we agree that users’ visual preferences may vary across different image content (e.g., topics and scenarios). When constructing our personalized preference dataset, we also take into account the diversity of users’ tastes by assigning multi-preference profiles to 20K users. This ensures that each user’s reference images reflect their varied personalized preferences.
>
> Furthermore, your question inspires us to adopt a more structured elementary preference profile to capture content-dependent personalized preferences. Specifically, we can introduce “content” as a key categorical element within our current preference profiles. By binding specific visual preferences to content categories, we can construct content-aware personalized preference profiles. For example, if a user exhibits different visual preferences for *landscape* and *building* content, the resulting profile for this user would be structured as follows:
>
> | **Content** | **Art style** | **Color** | **Art Medium** | **Saturation** | **Detail** |
> | :--- | :---: | :---: | :---: | :---: | :---: |
> | _**Landscape**_ | Ceramic Art | Harvest Gold | Spray Paint | Intense | Intricate |
> | _**Building**_ | Manga | Cranberry Red | Colored Pencil | Vibrant | Defined |
>
> By building a dataset based on such content-aware personalized preference profiles and training model on it, the model can uncover the relationship between users’ visual attribute preferences and content, enabling it to capture subtle real user preferences at a finer granularity.

---

> ### Author Response · Authors · 2025-11-20
>
> **[Q2]: The motivation for using element-wise preference profiles instead of a continuous latent user-preference representation.**
>
> **[A]:** The motivation for selecting element-based preference profiles mainly lies in two aspects:
>
> **1.  Address data scarcity by bridging various users.** A core challenge in personalized preference assessment is that each user’s personalized data are scarce and not easily scalable. To address this, we introduce a common preference profile composed of multiple visual elements to bridge various users. Although each user’s personalized preferences are unique, the fundamental elements that form them can be shared and transferred across users. This allows the model to leverage large-scale user samples for training to effectively predict preference profiles, thereby mitigating the issue of limited personalized data.
>
> In contrast, with continuous latent representations, each user’s preference embedding is independent and lacks explicit connections, making it difficult for the model to transfer knowledge across users and to perform large-scale training when each user has limited personalized data.
>
> **2. Achieve interpretable assessment across multiple dimensions.** Our work aims to build a reasoning-based assessment system, rather than simply producing a final result. The element-wise preference profile consists of human-interpretable visual elements, enabling a visual language model (VLM) to perform CoT-style reasoning based on it. As shown in Figure 2, our method can leverage the predicted profiles to provide multi-dimensional, interpretable scoring of candidate images.
>
>  In contrast, continuous latent representations act like a black box: although they can capture user preferences at the feature level, they are difficult to use for producing interpretable assessment.

---

> ### Author Response · Authors · 2025-11-27
> **Looking forward to further discussions**
>
> Dear Reviewer oEnc,
>
> We sincerely thank you for your insightful comments on our paper, and we genuinely hope that our response could address your concerns. As the discussion period is nearing its end with *less than one week remaining*, we look forward to receiving your feedback. Please feel free to contact us if you have any further inquiries.
>
> Thank you once again for your valuable contributions to enhancing the quality of our work.
>
> Authors of Submission 3233

---

### Official Review · Reviewer_2Qg5 · 2025-10-29

**Soundness:** 3
**Presentation:** 4
**Contribution:** 3
**Rating:** 6
**Confidence:** 4

**Summary:**

This paper presents PreferThinker, a reasoning-based personalized image preference assessment system. The key idea is to predict a preference profile to bridge various uses, allowing large-scale user data to be leveraged for training profile prediction and capturing complex
personalized preferences. To this end, a CoT-style dataset annotated with preference profiles and high-quality reasoning for interpretability supervision is constructed. A two-stage a two-stage training strategy comparing Cold-start SFT and reinforcement learning is utilized to enpower the model with reasoning capabilities. Experiments on the constructed dataset shows that the proposed method outperforms existing approaches.

**Strengths:**

This paper addresses personalized image preference assessment from a novel visual preference profile based interpretable perspective.

A large-scale Chain-of-Thought (CoT)-style personalized assessment dataset annotated with diverse user preference profiles and high-quality CoT-style reasoning is contructed, enabling explicit supervision of structured reasoning.

Experiments demonstrate the superiority of the proposed method.

**Weaknesses:**

My main concerns is how the proposed method generalizes to real-world images. To costruct a large-scale CoT-style dataset that provides high-quality reasoning supervision, the authors propose to combine several random profiles with initial prompts and feed into a text-to-image model to generate each user’s reference images (preferred and non-preferred) and two candidate images. However, the generated images ,as shown in paper and supplementary, lack photorealism, and would also cover a very limtied range of categories. Though the experiments demonstrate the superiority of the proposed method, the main experiments are conducted on the collected dataset, failing to validate the generalization to real-world images.

On the other hand, it is still unclear to me whether the compared methods in paper are trained or fine-tuned on the collected dataset for fair comparison of assessment accuracy. Moreover, the evaluation on the PickaPic dataset is somehow confusion to me. As described in paper, the experiment on PickaPic reflects general preferences, rather than personalized preference. However, the results in Table 1 show that the proposed method ranks second on unseen PickaPic data for general preference assessment, this makes me doubt whether the proposed method can indeed extract and understand personalized visual preference profile.

Currently, the employed visual preference profile consists of five visual elements while the user study is conducted with 15 visual elements. Therefore, I would like to see experimental analysis on how the number of visual elements in a visual preference profile affects the final results.

In fact, it is difficult for a person to rate his/her visual preference to an image using an absolute score, but it is easy for a person to perform relative preference comparison between two images. Hence, I suggest the authors to evaluate the proposed method on such preference ranking data.

**Questions:**

What is the limitation of the proposed method?

How does the effectiveness of the proposed method on personalized image generation?

In my opinion, the current 5-element visual preference profile mainly summarizes the visual tone of an image, I wonder whether it is possible to make the model learn to find the visual elements that determines his/her visual preference?

---

> ### Author Response · Authors · 2025-11-20
>
> Dear Reviewer 2Qg5, we thank you for your valuable comments and suggestions. We appreciate your insightful questions and hope our responses could address your concerns.
>
> **[W1-1]: Personalized preference assessment on real-world images.**
>
> **[A1-1]:** Thanks for your thoughtful question. personalized preferences assessment for real-world images is indeed meaningful, as humans may have distinct preferences for images produced with different photographic styles (e.g., composition, shooting techniques, and content). However, our current work primarily focuses on personalized preferences assessment for generated images and has not yet extend to real-world imagery.
>
> **[W1-2]: The generated images exhibit limited photorealism and category diversity.**
>
> **[A1-2]:** The limited photorealism of the generated images is primarily constrained by the capabilities of the generative model used. The main purpose of employing this model in our study is to construct a dataset suitable for personalized preference assessment, rather than to generate highly realistic images. In future work, more advanced generative models can be adopted to further enhance the visual realism of the images.
>
> In constructing the dataset, we have carefully ensured diversity in both content and visual attributes: 190 K prompts are used to cover a wide range of content categories, and 288 visual attribute terms are introduced to form diverse profiles. However, our dataset does not yet include real-world images. Future work could involve collecting personalized data for real images and exploring the key factors shaping users’ personalized preferences for real-world imagery.
>
> **[W2]: Whether the compared methods are trained on the collected dataset PreferImg-CoT?**
>
> **[A]:** We use the pre-trained models of the compared methods without training them on our PreferImg-CoT dataset. This is mainly because most of methods are designed for general preferences, and their architectures do not support incorporating users’ reference images for in-context learning. The only personalized assessment method, ViPer, similarly does not support profile prediction and CoT-style reasoning. Both closed-source and open-source MLLMs also do not require any additional training.
>
> To enable a more informative comparison, we train two representative methods on our dataset: ImageReward [1] for general preferences and ViPer [2] for personalized preferences. For ImageReward, the training utilizes candidate images, text prompts, and preference choices. For ViPer, the training utilizes reference images, candidate images, and preference choices. The results after training are shown in the table below. It is observed that ImageReward still struggles with personalized preferences since it does not consider users’ personalized reference images. Meanwhile, ViPer shows improved accuracy on PreferImg after training on our dataset, yet still underperforms compared to the proposed method.
>
> The symbol * indicates training on our dataset
> | Method | Seen-SP | Seen-MP | Unseen-SP | Unseen-MP | PickaPic |
> | :--- | :---: | :---: | :---: | :---: | :---: |
> | ImageReward | 52.2 | 51.2 | 49.0 | 57.2 | 60.7 |
> | ViPer | 92.4 | 78.0 | 93.4 | 80.0 | 62.2 |
> | ImageReward* | 53.0 | 50.8 | 51.6 | 50.0 | 49.8 |
> | ViPer* | 95.0 | 86.4 | 93.6 | 86.8 | 61.0 |
> | **PreferThinker (ours)** | **96.6** | **92.0** | **96.4** | **92.8** | **65.7** |
>
> [1] Jiazheng Xu, et al. ImageReward: Learning and Evaluating Human Preferences for Text-to-Image Generation. In NeurIPS 2023.
>
> [2] Sogand Salehi, et al. ViPer: Visual personalization of generative models via individual preference learning. In ECCV 2024.
>
> **[W3]: Performance of the proposed method on the unseen real-world Pick-a-Pic dataset that reflects general preferences.**
>
> **[A]:** Thanks for your careful review. Since there is no public dataset for real-world personalized preferences, we create a user-specific benchmark using the user ID in the Pick-a-Pic dataset to simulate personalized preference assessment. Specifically, we group samples by user ID, resulting in 894 user-specific samples. Each sample contains five pairs of reference images (preferred v.s. non-preferred) as prior information on the user’s preferences, along with one pair of images to be assessed.
> Although the labels of Pick-a-Pic reflect general preferences, each user’s reference images can partially reveal their unique preferences. Our method learns from these reference images to implicitly extract each user’s preference profile. Therefore, even though the Pick-a-Pic data are unseen and the labels reflect general preferences, our method still achieves second-best performance, demonstrating its ability to generalize to real users.

---

> ### Author Response · Authors · 2025-11-20
>
> **[W4]: Impact of the number of visual elements on final results.**
>
> **[A]:** We modify the question prompt to vary the number of visual elements included in the predicted profiles during inference and study how this affects final results. The results on real Pick-a-Pic dataset are shown in the table below. We can observe that when the number of visual elements ranges from 1 to 5, accuracy increases as more elements are included, indicating that a richer preference profile better captures users’ personalized preferences.
>
> ***C: Color, A: Art style, M: Art medium, S: Saturation, D: Detail***
>
> | **Num.** | **1 (_C_)** | **2 (_C+A_)** | **3 (_C+A+M_)** | **4 (_C+A+M+S_)** | **5 (_C+A+M+S+D_)** |
> | :--- | :---: | :---: | :---: | :---: | :---: |
> | **Acc.** | 62.2| 63.9 | 65.0| 65.2| 65.7 |
>
> Our current preference profile consists of five key visual elements. To study the impact of incorporating more visual elements on performance, it would be necessary to reconstruct new large-scale personalized profiles, and generate the corresponding image dataset along with  CoT-style reasoning annotations, which is a highly time-consuming process. Given the limited time during the rebuttal phase, we plan to design preference profiles with additional visual elements and construct corresponding datasets for further analysis in future work.
>
> **[W5]: Suggestion for evaluation on preference ranking data.**
>
> **[A]:** Thanks for your valuable suggestion. We agree that humans can more easily compare the relative preference between two images than rate an image, this is precisely the approach we adopt. In our work, the labels for the candidate images are based on such relative preferences: both our PreferImg dataset and the Pick-a-Pic dataset provide labels indicating which image a user prefers in a pair. Our model assesses candidate images through in the same way. Although our method produces multi-dimensional scores for interpretability, the final preference is determined by computing and comparing the total scores of images A and B.
>
> The motivation for multi-dimensional scoring is to provide interpretability: we not only want the model to predict which image a user prefers, but also to explain the reasoning behind the choice. By generating scores across multiple dimensions, the model reveals the strength of a user’s preference for each aspect of an image, making the assessment more interpretable.
>
> **[Q1]: What is the limitation of the proposed method?**
>
> **[A]:**  The main limitation of our method is that the personalized preference dataset built on synthetic profiles may not fully capture the complex and subtle real human preferences. Our current preference profile only contains five key visual attributes that influence human visual preferences. However, real users’ preferences are often more complex and subtle, shaped by multiple factors. Beyond visual attributes, they are also closely related to image content semantics and users’ backgrounds (e.g., age, gender and culture). Consequently, our method may struggle with complex and hard-to-articulate real user preferences, as discussed in Section E.4 in Appendix and Figure S22. In future work, we will incorporate additional key factors into preference profiles to more comprehensively capture real users’ preferences.
>
> **[Q2]: How does the effectiveness of the proposed method on personalized image generation?**
>
> **[A]:** Our method can predict a user’s preference profile from their personalized reference images. We then use the predicted profile to optimize the initial prompt, and the optimized prompt is fed into a text-to-image model to generate personalized images. As shown in Figure 6 of the main paper and Figure S21 in the appendix, the visual tone of the generated images is strongly aligned with that of the user’s personalized reference images, demonstrating the effectiveness of our method in facilitating personalized image generation.

---

> ### Author Response · Authors · 2025-11-20
>
> **[Q3]: Can the model learn to adaptively identify the specific visual elements that determine a user's preference?**
>
> **[A]:** Thanks for your insightful question. We agree that adaptively identifying the key visual elements that shape user preferences is a highly meaningful, especially when dealing with subtle and hard-to-articulate real user preferences. This insight inspires us to consider a new approach of constructing preference profiles. We can construct a real personalized dataset with “pseudo” preference profile labels. Specifically, we first collect real users’ personalized data, including user IDs, reference images, candidate images and preference selections. Then, we can leverage the strong image understanding capabilities of advanced closed-source models (e.g., GPT-5) to extract pseudo preference profile labels from users’ reference images and   adaptively identify the key factors that determine user preferences. By building a dataset based on these pseudo preference profiles and training model on it, the model can adaptively identify the key visual elements driving user preferences and more flexibly captures subtle real user preferences.

---

> ### Author Response · Authors · 2025-11-27
> **Looking forward to further discussions**
>
> Dear Reviewer 2Qg5,
>
> We sincerely thank you for your insightful comments on our paper, and we genuinely hope that our response could address your concerns. As the discussion period is nearing its end with *less than one week remaining*, we look forward to receiving your feedback. Please feel free to contact us if you have any further inquiries.
>
> Thank you once again for your valuable contributions to enhancing the quality of our work.
>
> Authors of Submission 3233

---

> > ### Comment · Reviewer_2Qg5 · 2025-11-28
> >
> > I thank the authors for the detailed rebuttal. Most of my concerns are addressed, so I will raise my rating.

---

> > > ### Author Response · Authors · 2025-11-28
> > >
> > > Thanks for your valuable suggestions and positive feedback on our work! They are very helpful for us to improve the paper.

---

### Official Review · Reviewer_PeqQ · 2025-10-29

**Soundness:** 3
**Presentation:** 3
**Contribution:** 3
**Rating:** 8
**Confidence:** 2

**Summary:**

This paper introduces PreferThinker, a new framework for personalized image preference assessment.
The authors pointed out that currently there is a lack of user-specific data.
To overcome this issue, the paper proposes a common preference profile for bridging various users.
This profile enables the model to leverage large-scale data to learn user preferences.
The PreferThinker framework operates on a "predict-then-assess" paradigm: It first predicts a user's visual preference and non-preference profiles based on a small set of reference images. Then, it uses this predicted profile as a criterion to generate interpretable, multi-dimensional scores and a Chain-of-Thought (CoT) assessment for candidate images.

**Strengths:**

This paper tries to tackle the problem of personalized preference assessment by using the idea of common preference profile as a bridge between users. This idea is novel. Beyond this, the paper also introduces a new, large scale dataset for personalized assessment. The experiments are robust, covering seen vs. unseen profiles , single vs. multi-preference users , and robustness to the number of reference images.

**Weaknesses:**

The primary weakness is that the main dataset, PreferImg-CoT, is built on simulated user preferences. While the simulation pipeline is well-designed (based on a user study to find 5 key elements ), simulated profiles may not capture the full, complex, and sometimes contradictory or hard-to-articulate nature of real human preferences.

**Questions:**

Given the primary limitation is the simulated dataset, could you discuss the feasibility of collecting a (perhaps smaller) "gold standard" test set with real personalized data?

---

> ### Author Response · Authors · 2025-11-20
>
> Dear Reviewer PeqQ, we thank you for your insightful comments. We provide our replies to the comments below, and hope they could address your concerns.
>
> **[W1]: Dataset build on simulated profiles may not fully capture complex real user preferences.**
>
> **[A]:** Thanks for your careful review. We agree that our dataset built on simulated profiles cannot fully capture the complex real user preferences. One main reason for using a simulated dataset is that our proposed “predict-then-assess” CoT-style paradigm requires a dataset annotated with both preference profiles and CoT-style reasoning, which is unavailable in real datasets.
>
> Nevertheless, after being trained on our dataset, our method achieves the second-best performance on the unseen real dataset Pick-a-Pic, surpassing most existing methods, including the state-of-the-art closed-source MLLM. Notably, the first-place method PickScore is trained on Pick-a-Pic, whereas our method does not use this dataset for training, revealing its generalization to real users. We also agree that this generalization is limited, and the model may still fail when faced with more complex and subtle real user preferences. This limitation is discussed in detail in Section E.4 in Appendix and Figure S22.
>
> To better handle complex real users’ personalized preferences, we propose two potential directions for future work:
>
> **1. Constructing a more comprehensive personalized preference profile.** Our current preference profile only contains five key visual attributes that influence human visual preferences. However, real users’ preferences are often complex and subtle, shaped by multiple factors. Beyond visual attributes, they are also closely related to image content and users’ backgrounds (e.g., age, gender and culture).  In future work, we will incorporate additional factors into preference profiles to more comprehensively capture real users’ preferences.
>
> **2. Generating pseudo-labels of personalized preference profiles for real users.** we can construct a real personalized preference dataset with “pseudo” preference profile labels. Specifically, we first collect real users’ personalized data, including user IDs, reference images, candidate images and preference choices. We then leverage the strong image understanding capabilities of advanced closed-source MLLMs (e.g., GPT-5.1) to extract “pseudo” preference profile labels from the users’ reference images and adaptively identify the key factors that determine user preferences. Based on this dataset, the model trained using it can more flexibly capture subtle real users’ preferences.
>
> **[Q1]: The feasibility of collecting a "gold standard" test set with real personalized data.**
>
> **[A]:** Thanks for your insightful question. We believe it is feasible to collect a real personalized preference test set through online platforms, similar to how Pick-a-Pic [1] and HPS [2] were built using platform-based user data to construct real general preference dataset. A possible collection pipeline is outlined as follows and consists of five main steps:
>
> **Step 1: Real user recruitment**
>
> First, we can recruit a group of real users with high reliability and diverse backgrounds through online platforms and assign each of them a unique user ID.
>
> **Step 2: Reference images collection**
>
> Then, we ask each user to upload a set of personalized reference images, consisting of multiple pairs of preferred and non-preferred images generated from the same prompt.
>
> **Step 3: User and reference image filtering**
>
> To ensure data quality, we can further manually review all collected reference images and remove any unusable submissions or users who provide invalid content.
>
> **Step 4: Candidate image assessment**
>
> Next, we ask the filtered users and ask them to make preference choices (“which one do you prefer?”) on a series of new candidate image pairs.
>
> **Step 5: Construction of the real personalized preference test set**
>
> Finally, we aggregate all high-quality data and bind each user’s user ID, personalized reference images, candidate images, and corresponding choices to form the final real personalized preference test set.
>
> [1] Yuval Kirstain, et al. Pick-a-Pic: An Open Dataset of User Preferences for Text-to-Image Generation. In NeurIPS 2023.
>
> [2] Xiaoshi Wu, et al. Human Preference Score: Better Aligning Text-to-Image Models with Human Preference. In ICCV 2023.

---

> ### Author Response · Authors · 2025-11-27
> **Looking forward to further discussions**
>
> Dear Reviewer PeqQ,
>
> We sincerely thank you for your insightful comments on our paper, and we genuinely hope that our response could address your concerns. As the discussion period is nearing its end with *less than one week remaining*, we look forward to receiving your feedback. Please feel free to contact us if you have any further inquiries.
>
> Thank you once again for your valuable contributions to enhancing the quality of our work.
>
> Authors of Submission 3233

---

### Official Review · Reviewer_PbCi · 2025-11-01

**Soundness:** 3
**Presentation:** 2
**Contribution:** 2
**Rating:** 4
**Confidence:** 4

**Summary:**

This work proposes PreferThinker, a system for personalized image reward scoring with chain of thought reasoning to predict a user's preference profile. To train this system, the authors first construct PreferImg-CoT, using Claude 3.7 to generate a reasoning trace of each individual user's score given reference images. The authors use this dataset to train Qwen-2.5-VL-7B with SFT and GRPO, and evaluate on their PreferImg dataset and Pick-a-Pic v1.

**Strengths:**

* This work proposes an interesting system to predict a user's preference profile for text-to-image generation, and then score generated images for that individual
* This work contributes a new large scale synthetic preference dataset PreferImg created from 80K synthetic user preference profiles with attributes that the authors choose after a real-world user study. This dataset can be a valuable resource for the text-to-image reward modeling community
* The authors demonstrate that their dataset can learn rewards that generalize well when new users have similar preference profiles to users seen during training (Tab 1) via cold start SFT and RL training of Qwen 2.5 VL

**Weaknesses:**

* The authors argue that *"although each user’s personalized preferences are unique, the key visual elements that shape these preferences are shared"* (L197-198), and they mention discrete attributes that users rank highly as important to them (art style, color, detail, art medium and saturation). I feel that this a strong assumption to make - what about individual preference differences that are more semantic in nature for a given prompt? Is it possible to discretize real-world user preferences of generated images in this manner?
* **My primary concern**: PreferThinker appears robust to unseen users when the distribution of their preference profiles is shared with seen users (I.e. they are both sampled from PreferImg). How robust is PreferThinker to preferences outside this distribution? Since PreferImg is constructed with 80K synthetic preference profiles, it is unclear to me how robust these preference profiles will be to real-world datasets. While the authors do evaluate on Pick-a-Pic, PreferThinker's accuracy (67%) does not seem compelling when compared to extremely lightweight pluralistic reward modeling baseline [1], which gets 71% accuracy on V1 and 70.5% on V2 (no-leakage) with just 6M trainable parameters
* The presentation has room for improvement, especially the figures, which are quite small, dense, and difficult to read (especially Figure 8). Figure 1 and 2 can be made bigger and Fig 3 moved to the Appendix. It is very hard to parse Table 2 and 3 as their captions are nearly contiguous.

[1] Chen et al., "PAL: Sample-Efficient Personalized Reward Modeling for Pluralistic Alignment", ICLR 2025.

**Questions:**

* Which version of Pick-a-pic is used in your experiments, v1 or v2?

---

> ### Author Response · Authors · 2025-11-20
>
> Dear Reviewer PbCi, we thank you for your valuable comments and suggestions. We appreciate your insightful questions and hope our responses could address your concerns.
>
> **[W1-1]:**  **Considering only visual attribute preferences is a strong assumption-what about semantic preferences?**
>
> **[A1-1]:** We agree that differences in individual preferences exist not only in visual attributes but also in the semantics of the prompt, which are related to image content. In this work, we mainly focus on individual preferences for visual tone, formed by multiple visual attributes, given a prompt with fixed semantics. Incorporating both visual attributes and semantics preferences into preference profiles would more comprehensively capture real individual preferences and facilitate the exploration of the relationship between content semantics and visual attributes, as humans may have inconsistent visual preferences across different content. Moreover, real users’ preferences are often complex and subtle, shaped by multiple factors. Beyond visual attributes and content semantics, they are also closely related to users’ backgrounds (e.g., age, gender and culture). In future work, we will incorporate additional factors into preference profiles to more fully capture real users’ preferences.
>
> **[W1-2]: Is it possible to use preference profile to discretize real-world user preferences?**
>
> **[A1-2]:** Yes, we believe it is possible. The proposed profile has been preliminarily validated for its effectiveness.  Table 1 shows that our method achieves second-best performance on the unseen real-world dataset Pick-a-Pic, outperforming most methods, including the state-of-the-art closed-source MLLM. Notably, the first-place method, PickScore, is trained on the Pick-a-Pic dataset, whereas our approach does not use this dataset for training, demonstrating the generalization ability to real users.
> However, our current preference profile may not fully capture the complex and subtle real user preferences. As noted in our response [A1-1], real users’ preferences are often shaped by multiple factors. To address this issue, two approaches can be considered. First, as noted in our response [A1-1], we can extend additional factors into the profiles to more fully capture real users’ preferences.
>
> Second, we can construct a real personalized preference dataset with “pseudo” preference profile labels. Specifically, we first collect real users’ personalized data, including user IDs, reference images, candidate images and preference choices. We then leverage the strong image understanding capabilities of advanced closed-source MLLMs (e.g., GPT-5) to extract “pseudo” preference profile labels from the users’ reference images and adaptively identify the key factors that determine user preferences. Based on this dataset, the model trained using it can more flexibly capture subtle real users’ preferences.
>
> **[W2-1]: How robust is the proposed method on real-world dataset?**
>
> **[A2-1]:** Since no public dataset exists for real-world personalized preferences, we construct a user-specific benchmark by leveraging the user ID in the real-world Pick-a-Pic dataset to simulate personalized preference assessment. Specifically, we group samples according to user ID, resulting in 894 user-specific samples. Each sample includes five pairs of reference images (preferred vs. non-preferred) that serve as prior information on the user’s preferences, along with one pair of images to be assessed.
> As noted in our response [A1-2], our method achieves the second-best performance on the unseen Pick-a-Pic dataset, surpassing most existing methods. It should be noted that the first-place method, PickScore, is trained on the Pick-a-Pic, whereas our method does not use this dataset for training, demonstrating its generalization to real users.

---

> ### Author Response · Authors · 2025-11-20
>
> **[W2-2]: Comparison of our accuracy on Pick-a-Pic with the accuracy reported by the PAL paper on Pick-a-Pic.**
>
> **[A2-2]:**  Our training and testing setup on Pick-a-Pick dataset differs from that of PAL, so the accuracy results are not directly comparable. Differences can be summarized in two aspects:
>
> **1. Different training setup:** Our model is trained solely on our dataset, PreferImg-CoT, and then evaluated on an unseen user-specific Pick-a-Pic benchmark. We do not use Pick-a-Pic for training, as it lacks the preference profiles required for our method. In contrast, as reported in the Table 2 of PAL paper [1], PAL is trained on the Pick-a-Pic training set and evaluated on its test set.
>
> **2. Different testing setup and benchmark:** Our task focuses personalized image preference assessment, which requires each user’s reference images as prior information. As noted in our response [A2-1], since no real dataset for personalized preferences exists, we construct a user-specific Pick-a-Pic benchmark based on user IDs. Therefore, our testing setup and benchmark differ from those of PAL. We have cited this work that focuses on lightweight reward model in our revised paper.
>
> [1] Chen et al., PAL: Sample-Efficient Personalized Reward Modeling for Pluralistic Alignment, In ICLR 2025.
>
> **[W3]: Advice about the improvement of the presentation.**
>
> **[A]:** Sincerely thanks for your valuable advice. We have improved the presentation of the paper based on your feedback, and all changes in the revised manuscript are highlighted in light blue, as summarized below:
>
> 1. The figures in the paper have been improved for better readability, particularly Figure 1, Figure 2, and Figure 5.
>
> 2. Tables 2 and 3 have been revised to make their captions more distinct.
>
> 3. Figure 7 (previously Figure 8) has been modified for easier reading.
>
> 4. Figure 3 has been moved to the appendix.
>
> **[Q1]: The version of Pick-a-Pic we used in our experiments.**
>
> **[A]:** The version of Pick-a-Pic we used is v2.

---

> ### Author Response · Authors · 2025-11-27
> **Looking forward to further discussions**
>
> Dear Reviewer PbCi,
>
> We sincerely thank you for your insightful comments on our paper, and we genuinely hope that our response could address your concerns. As the discussion period is nearing its end with *less than one week remaining*, we look forward to receiving your feedback. Please feel free to contact us if you have any further inquiries.
>
> Thank you once again for your valuable contributions to enhancing the quality of our work.
>
> Authors of Submission 3233

---

### Author Response · Authors · 2025-11-13
**Add citations in two place**

We sincerely thank all the reviewers for their valuable suggestions and careful review.
The first author of [1] contacted us and requested us to cite their paper. Consequently, we have revised our manuscript to incorporate this citation. The modifications have been highlighted in yellow within the text, and the explanation is detailed in the footnote. We also offer further clarification below.

**1. Added Citation at Location 1 (Line 238):**

We added the citation: …inspired by [1] …

**2. Added Citation at Location 2 (Line 245):**

We added the citation: …follow the PrefGen [1] to select 190 K prompts…

**Our Clarification:**

This work is supported by a joint project by a company and a university. The text prompts we used were provided by the company when the second author of our paper was a research intern in the company. The first and second authors of our paper knew that the first author of [1] contributed to the collection of these text prompts, but believed the ownership of the prompts belonged to the company. As requested by the first author of [1], after coordinating with the company and the first author of [1], we cited [1] here to emphasize the source of the text prompts.

We will continue to carefully respond to each reviewer's comments in the coming days.

[1] PrefGen: Multimodal Preference Learning for Preference-Conditioned Image Generation, https://openreview.net/pdf?id=8iGclsodrJ.

---

### Author Response · Authors · 2025-11-20

We first thank the reviewers for affirming our main contributions: an **interesting** [PbCi] **and carefully designed** [oEnc] **framework** that provides **better modularity for image generation and interpretability** [oEnc]; a **novel common visual preference profile** [PeqQ, 2Qg5, oEnc] that serves as a bridge across users; a **valuable resource** [PbCi] for the community in terms of **new large-scale CoT-style dataset** [PbCi, PeqQ, 2Qg5]; and **robust experiments** [PeqQ] demonstrating **superior performance** [2Qg5, oEnc].

As for the main problems, most reviewers [PbCi, PeqQ, oEnc] are particularly concerned with capturing complex and subtle personalized preferences of real users. In addition, reviewer PbCi is mainly concerned with the robustness of proposed method to real-world preferences, and   reviewer PeqQ cares about the collection of real user test set with personalized data, and reviewer 2Qg5 mainly focuses on the performance on real-world images, and reviewer oEnc questions the motivation for using elementary preference profiles rather than continuous latent representations.

To sum up, we deeply thank the reviewers for their appreciation of our contributions, and for the valuable suggestions that advance the integrity of this work.

---

### Meta-Review · Area_Chair_EduG · 2026-01-09

**Summary:**

This paper proposes PreferThinker, a reasoning-based framework for personalized image preference assessment. Reviewers’ core concerns focus on three aspects: 1) generalization limitations due to reliance on simulated preference profiles in PreferImg-CoT, and doubts about its ability to capture complex real-world user preferences; 2) rationality of method design, including lack of coverage of semantic preferences.

**Reviewer Concerns:**

Addressed concerns: 1) The unfair comparison issue with the PAL baseline was clarified by the authors; 2) Poor figure readability was resolved; 3 The impact of the number of visual elements in preference profiles was verified by additional experiments; 4) The effectiveness of personalized assessment was further validated by constructing a user-specific benchmark using Pick-a-Pic user IDs.

Outstanding concerns: 1) Simulated datasets still cannot fully capture the subtle nature of real human preferences; 2. The model has not been validated on real-world images; 3) The current preference profile lacks integration of semantic preferences and user background factors.

**Reviewer Scores:**

PbCi: Most concerns were partially addressed, with the performance comparison ambiguity and visualization issues resolved.

PeqQ: The author’s response fully addressed the feasibility of constructing a real test set and acknowledged dataset limitations.

2Qg5: The reviewer explicitly stated that most concerns were addressed and intended to raise the score.

oEnc: The author clarified the motivation for using discrete profiles and proposed potential optimization directions for content-aware profiles.

---

### Decision · Program_Chairs · 2026-01-26

Accept (Poster)